# DNA binding and RAD51 engagement by the BRCA2 C-terminus orchestrate DNA repair and replication fork preservation

Youngho Kwon [1,11], Heike Rösner [2,11], Weixing Zhao[1], Platon Selemenakis [3,7], Zhuoling He[1], Ajinkya S. Kawale[1,8], Jeffrey N. Katz[1], Cody M. Rogers[1], Francisco E. Neal [1], Aida Badamchi Shabestari[1], Valdemaras Petrosius[2], Akhilesh K. Singh [4,9], Marina Z. Joel[4,10], Lucy Lu[4], Stephen P. Holloway[1], Sandeep Burma [1,5], Bipasha Mukherjee[5], Robert Hromas[6], Alexander Mazin[1], Claudia Wiese [3] ✉, Claus S. Sørensen [2] ✉ & Patrick Sung [1] ✉

The tumor suppressor BRCA2 participates in DNA double-strand break repair by RAD51-dependent homologous recombination and protects stressed DNA replication forks from nucleolytic attack. We demonstrate that the C-terminal Recombinase Binding (CTRB) region of BRCA2, encoded by gene exon 27, harbors a DNA binding activity. CTRB alone stimulates the DNA strand exchange activity of RAD51 and permits the utilization of RPA-coated ssDNA by RAD51 for strand exchange. Moreover, CTRB functionally synergizes with the Oligonucleotide Binding fold containing DNA binding domain and BRC4 repeat of BRCA2 in RPA-RAD51 exchange on ssDNA. Importantly, we show that the DNA binding and RAD51 interaction attributes of the CTRB are crucial for homologous recombination and protection of replication forks against MRE11-mediated attrition. Our findings shed light on the role of the CTRB region in genome repair, reveal remarkable functional plasticity of BRCA2, and help explain why deletion of *Brca2* exon 27 impacts upon embryonic lethality.

Homologous recombination (HR) is a conserved process that ensures the error-free repair of DNA double-strand DNA breaks (DSBs)[1–3]. HR proteins are also involved in the repair of collapsed DNA replication forks and preservation of stressed replication forks against nucleolytic attack[4,5]. As such, HR defects lead to genome destabilization, neoplastic transformation of cells, and disease, cancer in particular. Notably, germline and somatic mutations in HR genes, such as *BRCA1*

and *BRCA2*, lead to a genomic instability phenotype that promotes breast, ovarian, and other cancer types[2,6,7]. Emerging evidence shows that BRCA1 and BRCA2 proteins regulate the activity of other HR factors, such as the RAD51 recombinase[8–12].

During HR, a primary lesion, such as a DSB, undergoes 5′ strand resection that is mediated by several nucleases[13,14]. DNA end resection generates a 3′ tailed ssDNA template for the assembly of a helical

[1]Department of Biochemistry and Structural Biology and Greehey Children's Cancer Research Institute, University of Texas Health Science Center at San Antonio, San Antonio, TX 78229, USA. [2]Biotech Research and Innovation Centre, Faculty of Health and Medical Sciences, University of Copenhagen, DK-2200 Copenhagen N, Denmark. [3]Department of Environmental and Radiological Health Sciences, Colorado State University, Fort Collins, CO, USA. [4]Department of Molecular Biophysics and Biochemistry, Yale University School of Medicine, New Haven, CT, USA. [5]Department of Neurosurgery, University of Texas Health Science Center at San Antonio, San Antonio, TX 78229, USA. [6]Department of Medicine, University of Texas Health at San Antonio, 7703 Floyd Curl Drive, San Antonio, TX 78229, USA. [7]Present address: Department of Cancer Biology, University of Texas MD Anderson Cancer Center, Houston, TX, USA. [8]Present address: Massachusetts General Hospital Cancer Center, Harvard Medical School, Charlestown, MA 02129, USA. [9]Present address: GentiBio Inc., 150 Cambridgepark Dr, Cambridge, MA 02140, USA. [10]Present address: Johns Hopkins University School of Medicine, Baltimore, MD 21205, USA. [11]These authors contributed equally: Youngho Kwon, Heike Rösner. ✉e-mail: claudia.wiese@colostate.edu; claus.storgaard@bric.ku.dk; sungp@uthscsa.edu

filament of RAD51 crucial for downstream repair events[7,14]. Specifically, the RAD51-ssDNA nucleoprotein filament, often referred to as the presynaptic filament, engages duplex DNA and searches the latter for a DNA sequence homologous to that of the bound ssDNA[15]. Upon the location of DNA homology, the presynaptic filament catalyzes the formation of a DNA joint between the bound DNA molecules. The RAD51-mediated DNA strand invasion and exchange reaction is enhanced by HR factors such as the tumor suppressor complex BRCA1-BARD1[7,9,16].

The assembly of the RAD51 presynaptic filament is the main rate-limiting step in HR execution. Extensive cytological and biochemical evidence shows that the assembly process is prone to interference by the heterotrimeric ssDNA binding protein RPA[10–12,17,18]. Several HR factors, known as recombination mediators, facilitate RPA-RAD51 exchange on ssDNA to enable the timely assembly of the presynaptic filament. These mediators include yeast Rad52, Brh2 in *Ustilago maydis*, and the tumor suppressor BRCA2 in mammals[18–20].

We are interested in defining the mechanism of BRCA2, an exceptionally large protein of 3,418 amino acid residues (384 kDa), in its HR mediator activity (Fig. 1a). BRCA2 possesses a DNA binding domain (DBD) that harbors three Oligonucleotide Binding (OB) folds and has specificity for ssDNA. BRCA2 also physically interacts with RAD51 through eight BRC repeats (BRC1 through BRC8)[21–23] and its C-terminal Recombinase Binding region (henceforth referred to as CTRB)[7,18,19,24]. BRC repeats serve distinct functions, namely, to recruit RAD51 to ssDNA, attenuate the affinity of RAD51 for dsDNA, and stabilize the presynaptic filament[23,25,26]. Unlike yeast Rad52, which physically interacts with RPA when executing its mediator function, BRCA2 is devoid of RPA interacting ability[10]. Interestingly, the BRCA2-associated DSS1 protein, which comprises just 70 amino acid residues and is highly acidic, provides the RPA interaction interface[10,27].

Little molecular information is available as to the mechanistic role of the CTRB in BRCA2 functions. CTRB is encoded by the last gene exon (exon 27 in humans) and its deletion engenders an HR defect, severe DNA damage sensitivity, and imparts elevated cancer risk and embryonic lethality in mice[28–32]. In isolation, CTRB interacts with the oligomeric form of RAD51, and CTRB mutations, (e.g., S3291A or E), weaken the affinity for RAD51. However, while these mutations compromise the protection of stressed DNA replication forks against nuclease attack, they have little impact on HR efficiency[30–35]. Since exon 27 deletion engenders phenotypic consequences much more severe than the S3291A mutation[29,31,32], it seems clear that CTRB possesses at least one other functional attribute needed for DSB repair and replication fork preservation.

Here, by employing a combination of bioinformatic, biochemical, biophysical, and cell-biological analyses, we demonstrate that CTRB harbors a linear motif that binds both ssDNA and dsDNA. Importantly, we show that, in isolation, the CTRB region alone (1) stimulates the recombinase activity of RAD51 and (2) allows RAD51 to utilize RPA-coated ssDNA as a template for DNA strand exchange. Moreover, we furnish evidence that the CTRB acts in synergy with the BRC repeats and DBD of BRCA2 in delivering RAD51 to ssDNA and RPA-coated ssDNA to seed presynaptic filament assembly. Through isolation of separation-of-function CTRB mutants differentially impaired for DNA binding or RAD51 interaction, we provide biochemical evidence that both CTRB activities contribute toward RAD51 enhancement and HR mediator activity. Cellularly, both CTRB attributes are indispensable for HR, resistance to Mitomycin C (MMC), Hydroxyurea (HU), and the PARP inhibitor Rucaparib, assembly of DNA damage-induced RAD51 foci, and the protection of stressed replication forks against MRE11-mediated nucleolytic degradation. Our results thus shed mechanistic light on a BRCA2 domain that helps orchestrate DNA damage repair and replication fork protection.

## Results

### Bioinformatic analyses revealed conserved motifs within CTRB

In our effort to understand the biological role of the BRCA2 CTRB, we reasoned that highly conserved regions in BRCA2 orthologs could indicate functionally important regions. Thus, we first employed sequence alignment of 63 BRCA2 orthologs across all kingdoms of eukarya to identify conserved regions in these proteins. Confirming previous reports[21,27], our analysis showed that BRCA2 orthologs exhibit a low degree of overall amino acid conservation and are challenging to align due to a vast number of indels, except for moderately conserved regions corresponding to the BRC repeats and the DBD. For simplification, we show only the alignment score for vertebrates (Fig. 1a: plot of the degree of conservation in vertebrates).

CTRB in human BRCA2 first appeared in its entirety in vertebrates (and in a shorter form within the kingdom of Animalia) and exhibits only a moderate identity score (Fig. 1a). However, when the percent identities of representative human BRCA2 domains, namely, a disordered region (residues 296–387), the BRC4 repeat (residues 1517–1548), the OB2-appended Tower domain (residues 2846–2950) OB-fold 1 (residues 2718–2800), and a CTRB fragment (residues 3260–3337) were plotted according to evolutionary divergence (Fig. 1b), we found that CTRB (3260–3337) is as stringently conserved as the BRC4 repeat or the Tower domain. Since BRC4 and the Tower domain are hallmark features among distant relatives of vertebrate BRCA2, we speculated that CTRB may possess another function.

We analyzed the CTRB fragment for folded versus disordered regions using various predictors for unbiased comparison[36–40] (Supplementary Fig. 1a). As previously predicted[21,27,41], the CTRB fragment appears to be moderate to largely disordered depending on the algorithm used. While MFDp2 predicted the entire fragment to be mainly unstructured, the remaining three predictors converged in predicting the region spanning residues 3270–3320 as only moderately disordered. It is possible that the region in question might undergo disorder-to-order transitions upon interaction with potential binding partners. We extended the architectural analysis to include the detection of MoRFs (molecular recognition features) and SLiMs (short linear motifs) to identify functional motifs. The analysis revealed that the conserved CTRB fragment contains multiple potential regulatory motifs, including possible phosphorylation sites, and also degradation and sumoylation motifs (Supplementary Table 1). The number and positioning of these short motifs vary greatly across different species. Nonetheless, two spikes of exceptionally high conservation are found within the disordered region. Firstly, we identified two tandem FXXP motifs starting at F3289 and F3298, respectively, which may confer RAD51 interaction capability in analogy to such RAD51 interaction motifs in Brh2, the *Ustilago maydis* BRCA2 ortholog[42]. Secondly, we discovered a highly conserved cluster of positively charged residues from K3263 to R3269 (Supplementary Fig. 1e).

### Biophysical evidence for a DNA binding activity in the CTRB

Next, we addressed the structure of the conserved CTRB fragment of BRCA2 containing the identified motifs and five potential CDK phosphorylation sites (amino acid residues 3260–3337) biophysically. Circular dichroism (CD) and $^{15}$N- Heteronuclear Single-Quantum Correlation Spectroscopy (HSQC)-NMR analyses revealed spectra characteristic of an intrinsically disordered species, being dominated by a random coil pattern (Supplementary Fig. 1b), and the$^{15}$N-HSQC NMR spectrum displayed a lack of chemical shift dispersion (Supplementary Fig. 1c). We continued to investigate possible residual or transient structure by secondary chemical shift analysis. We extracted the CA, CO, and NH secondary chemical shifts and analyzed them using the Secondary Structure Propensity (SSP) score as described in ref. [43] (Supplementary Fig. 1d). The SSP score analysis, together with the lack of $^{15}$N-HSQC-NMR peak dispersion, (Supplementary Fig. 1c), confirmed the extraordinarily extended nature of the region with just a

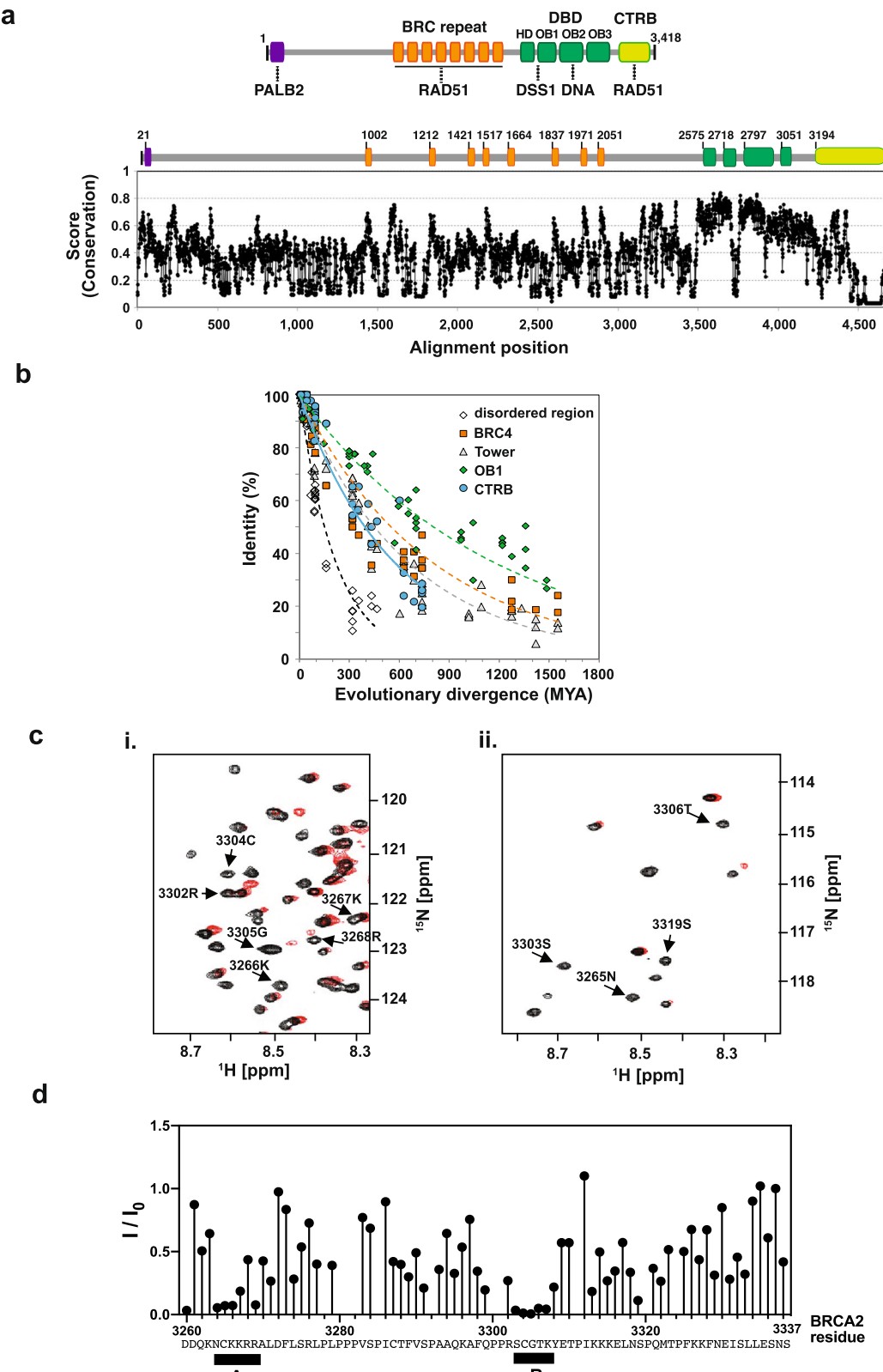

short stretch of residual helicity directly proceeding the second FXXP motif (Supplementary Fig. 1d). We note a region of apparent helicity near the N-terminus of the CTRB. It might result from an overestimation of helicity due to the N-terminal His6-tag. Although the CTRB core fragment under investigation is proline-rich, we saw no evidence for the formation of a PolyPro-II helix in the CD spectrum. Therefore, we concluded that the CTRB region adopts a conformation where motifs are aligned as beads on a string, in which the modularity of the motifs appears to be controlled by conserved prolines in key positions.

The CTRB harbors a highly conserved content of basic residues (pI= 9.1) organized in clusters (Supplementary Fig. 1e). This could indicate the possibility of a domain for either nucleic acid binding and/or interaction with negatively charged protein partners.

**Fig. 1 | Bioinformatic and NMR analyses of BRCA2 and the CTRB. a** (TOP) Schematic highlighting the location of different functional regions in BRCA2. (BOTTOM) The degree of conservation throughout full-length BRCA2 in selected vertebrates. Of note, the alignment score does not match the number of residues in human BRCA2 as a result of a substantial amount of indels in the BRCA2 sequence. For orientation, we have annotated the sequence position for human BRCA2 in the schematic on top of the conservation graph. Source data are provided as a Source Data file. **b** Evolutionary divergence for five different example regions of BRCA2: disordered: residues 296–387, BRC4 repeat: residues 1517–1548, OB1: residues 2718–2800, Tower domain: residues 2846–2950, CTRB core fragment: 3260–3337.

Source data are provided as a Source Data file. **c** $^1$H-$^{15}$N-HSQC HSQC of the CTRB polypeptide in the absence (black) or presence (red) of a 12-mer ssDNA. The $^1$H-$^{15}$N-HSQC NMR spectrum was recorded using 20 μM uniformly $^1$H-$^{15}$N-labeled BRCA2 CTRB (3260–3337) in 1 x PBS, pH 6.0, 10 mM DTT, at 5 °C. The full $^1$H-$^{15}$N-HSQC spectrum can be found in Supplementary Fig. 1c. **d** Relative change in peak intensities of the HSQC cross-peaks in the presence and absence of ssDNA. Two regions, designated A and B, where the effect of ssDNA is most prominent, are highlighted. The noise of the signals was ±5.5%. Source data are provided as a Source Data file.

Therefore, we investigated whether DNA would cause any perturbation in the $^{15}$N-HSQC-NMR spectrum (Fig. 1c, d). Importantly, upon adding a 12-nt ssDNA at an equimolar ratio, a number of CTRB residues displayed a significant decrease in the peak intensity ($I/I_0$), indicating a bipartite interaction modus with the DNA (Fig. 1c). The two clusters of residues displaying significant DNA-induced reduction in peak intensity are highlighted in Fig. 1d as Cluster A (residues 3264 to 3269) and Cluster B (residues 3303 to 3307). Importantly, biochemical testing, as documented below, provided corroborating evidence that the CTRB possesses a biologically relevant DNA binding attribute.

## Biochemical analysis of DNA binding by CTRB and mutant isolation

For biochemical studies, we expressed a His6-Thioredoxin-S tagged form of the CTRB (BRCA2 residues 3194–3418) in *E. coli* and purified it to near homogeneity (Fig. 2a, b). Purified CTRB was tested in the electrophoretic mobility shift assay (EMSA) with radiolabeled ssDNA, dsDNA, and DNA forks obtained by hybridizing oligonucleotides (Supplementary Table 2). Importantly, CTRB bound all three DNA species, showing the highest affinity for the fork substrate and about the same affinity for ssDNA and dsDNA (Fig. 2c and Supplementary Fig. 2a, b). Four basic CTRB amino acid residues ($^{3266}$KKRR) in Cluster A (Fig. 1d) were changed to alanine to generate the CTRB-4A allele. We could purify the mutant polypeptide to near homogeneity using the same procedure developed for the wild-type polypeptide (Fig. 2b). Importantly, EMSA revealed that the 4 A mutant being strongly impaired for DNA binding (Fig. 2c and Supplementary Fig. 2b). Affinity pulldown through the S-tag on the CTRB species verified that CTRB-4A is fully proficient in RAD51 interaction (Fig. 2d). Thus, the biochemical and NMR analyses (Figs. 1d, 2c and Supplementary Fig. 2b) are in strong congruence in identifying $^{3266}$KKRR in Cluster A as being crucial for DNA binding. Furthermore, affinity pulldown showed that the 4 A mutation has no negative impact on RAD51 interaction and thus represents a separation-of-function mutation (Fig. 2).

## Isolation of a CTRB mutant defective in RAD51 interaction

For addressing the role of RAD51 interaction in CTRB function, we constructed mutant variants that harbor either the S3291A or S3291E mutation, previously reported to disrupt RAD51 interaction[44]. Both CTRB mutants could be purified using the procedure that we developed for the wild-type counterpart. However, affinity pulldown revealed that CTRB-S3291A and CTRB-S3291E both retained significant, albeit reduced, an affinity for RAD51 (Supplementary Fig. 2c). In order to isolate a CTRB mutant that is defective in RAD51 interaction, we prepared four internally truncated mutant alleles of the TR2 region of the CTRB[44] that harbors the RAD51 interaction motif for affinity pulldown analysis. The results revealed that the deletion of residues 3296–3,300 or 3301–3305 abolishes RAD51 interaction, whereas the deletion of residues 3291–3295 or 3306–3310 has little or no impact on the affinity for RAD51 (Supplementary Fig. 2d). As identified in our bioinformatic analysis (Supplementary Fig. 1d), the portion of the CTRB (residues 3,296–3,305) crucial for RAD51 interaction (Supplementary Fig. 2d) harbors the second FXXP motif, which contains a

highly conserved PhePP motif, $^{3298}$FQPP. This PhePP motif resembles the WVPP motif in RAD51AP1 and the FVPP motif in BRCA2 that confer the ability to interact with the meiotic recombinase DMC1, a structural and functional homolog of RAD51, and also the FVTP motif in *U. maydis* Brh2 that is crucial for Rad51 interaction[42,45,46]. To test the relevance of the CTRB-FQPP motif in RAD51 interaction, we constructed the F3298A mutant alongside other mutants that are altered for nearby residues (namely, K3296A and R3302A) (Supplementary Fig. 2d). Testing of these mutants, as summarized in Supplementary Fig. 2d, revealed that only the F/A mutant is defective in RAD51 interaction (Fig. 2d). Importantly, by EMSA, we verified that the CTRB-F/A mutation has no impact on DNA binding (Fig. 2c and Supplementary Fig. 2a, b). Thus, the CTRB-F/A is a separation-of-function mutation that inactivates RAD51 interaction without affecting DNA binding.

## Enhancement of RAD51-mediated DNA strand exchange by CTRB

We employed a well-established oligonucleotide-based DNA strand exchange assay[47] to ask whether the CTRB would enhance the recombinase activity of RAD51. Importantly, the results showed that CTRB stimulates RAD51-mediated DNA strand exchange significantly (Fig. 2e). A control experiment confirmed that CTRB alone is devoid of DNA strand exchange activity (Fig. 2e, lane 7). We found that the RAD51 interaction defective CTRB-F/A and DNA binding impaired CTRB-4A mutants are much less capable of upregulating RAD51-mediated DNA strand exchange (Fig. 2g, lanes 7 and 10). Thus, CTRB enhances RAD51-mediated DNA strand exchange in a manner that is reliant on its RAD51 interaction and DNA binding attributes.

## HR mediator activity of CTRB in RAD51 presynaptic filament assembly

Next, we asked whether CTRB would enable RAD51 to utilize RPA-coated ssDNA as a template for DNA strand exchange. To test for such HR mediator activity, the ssDNA oligonucleotide used in DNA strand exchange was precoated with RPA and then incubated with RAD51 with or without an increasing amount of CTRB before incorporating the radiolabeled duplex DNA substrate (see Fig. 2f, i for schematic). As expected[10,11], precoating the ssDNA substrate with RPA led to strong inhibition of DNA strand exchange. Importantly, the results showed that CTRB proficiently enables RAD51 to utilize the RPA-coated DNA template to conduct DNA strand exchange (Fig. 2f). In contrast, neither the CTRB-F/A (RAD51 interaction defective) nor 4 A (DNA binding impaired) mutant showed significant HR mediator activity (Fig. 2g, lanes 8 and 11). Thus, CTRB represents an HR mediator module capable of enabling the utilization of RPA-coated ssDNA for RAD51 presynaptic filament assembly.

## Functional synergy of CTRB, BRC4, and DBD in RAD51 presynaptic filament assembly

Published work indicates that the BRC repeats and the OB-fold containing DBD together constitute a functional HR mediator module[10,12,18], and studies described above reveal such an HR mediator attribute in the CTRB as well (Fig. 2f). Given these findings, it became crucial to address whether the BRC repeats, DBD and CTRB would act

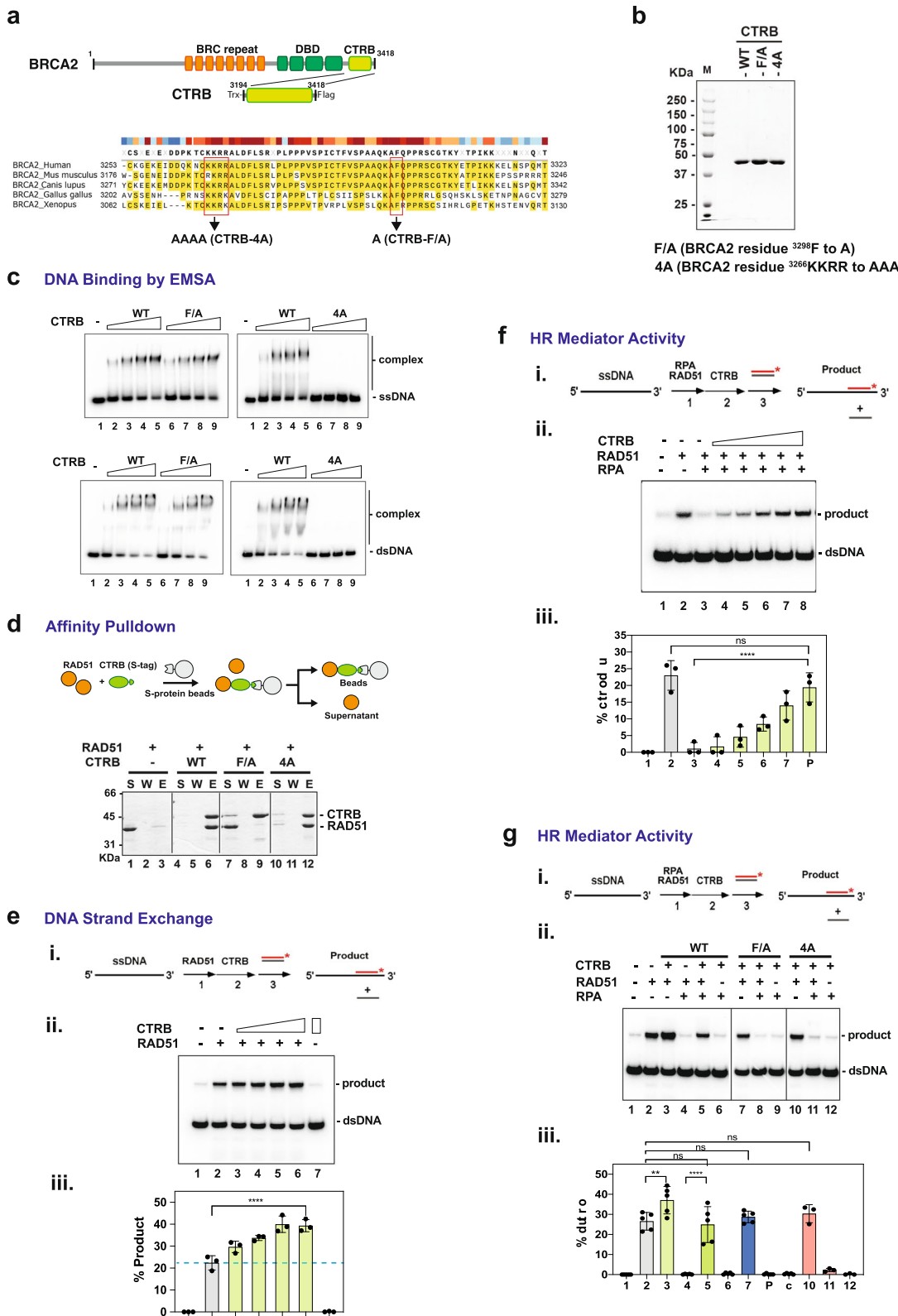

in synergy to nucleate RAD51 onto RPA-coated ssDNA to form a functional presynaptic filament. To this end, aside from the BRCA2 polypeptide comprising the BRC4 repeat, DBD, and CTRB (BRC4-DBD-CTRB) termed mini-BRCA2[10,12], we constructed additional BRCA2 polypeptides that harbor BRC4 and the DBD (BRC4-DBD) or the DBD and CTRB (DBD-CTRB) (Fig. 3a). We co-expressed these BRCA2-derived polypeptides with DSS1[10] and purified the resulting complexes

to near homogeneity (Fig. 3b). Importantly, in HR mediator test with RPA-coated ssDNA as a template, we observed that mini-BRCA2 is significantly more efficacious in DNA strand exchange restoration than BRC4-DBD or DBD-CTRB, with the latter two showing a comparable level of activity (Fig. 3c). These results show that BRC4, DBD, and CTRB functionally synergize in the delivery of RAD51 to RPA-coated ssDNA to mediate presynaptic filament assembly.

**Fig. 2 | HR mediator activity of the CTRB and relevance of RAD51 interaction and DNA binding. a** The CTRB region in BRCA2. Alignment of the domain in BRCA2 orthologs and the locations of the [3298]F to A and [3266]KKRR to AAAA mutations. **b** SDS-PAGE of purified wild-type CTRB, CTRB-F/A mutant, and CTRB-4A mutant. The experiment was repeated three times with similar results. Source data are provided as a Source Data file. **c** Binding of ssDNA and dsDNA by CTRB, CTRB-F/A, or CTRB-4A as examined by EMSA. Quantification of the DNA binding results is shown in Supplementary Fig. 2a. The samples were derived from the same experiment and the gels were processed in parallel. Source data are provided as a Source Data file. **d** S-tag pulldown for analysis of RAD51-CTRB interaction; RAD51 and CTRB (4 μg each) were tested. S: Supernatant containing unbound proteins, W: Wash, E: SDS-eluate. Note that the CTRB-F/A mutant is defective in RAD51 interaction, while the CTRB-4A mutant is proficient in this regard. The samples were derived from the same experiment and the gels were processed in parallel. The experiment was repeated three times with similar results. Source data are provided as the Source Data file. **e** DNA strand exchange assay with RAD51 and CTRB. (**i**) Reaction schematic and (**ii**) CTRB (lanes 3 to 6: 0.6, 1.2, 2.3, and 3 μM) was tested with ssDNA pre-incubated with RAD51 (2 μM). Lane 7 contained 3 μM CTRB and the

results were quantified and graphed in (**iii**). ($n = 3$ biologically independent experiments; mean ± SD; one-way ANOVA and Tukey's multiple comparison test; ****$p = 9.280 \times 10^{-6}$). Source data are provided as a Source Data file. **f** Testing of HR mediator activity of the CTRB. (**i**) Reaction schematic and (**ii**) CTRB (lanes 4 to 8: 0.4, 0.8,1.6, 2.4, 3 μM) was tested with RAD51 (2 μM) in the DNA strand exchange reaction with ssDNA precoated with RPA (600 nM) and the results were quantified and graphed in (**iii**). ($n = 3$ biologically independent experiments; mean ± SD; one-way ANOVA test and Tukey's multiple comparison test; ns (no significant difference): $p = 0.8620$; ****$p = 7.180 \times 10^{-5}$)). Source data are provided as a Source Data file. **g** The CTRB-F/A and CTRB-4A mutants (3 μM each) were examined as indicated. ($n = 5$ biologically independent experiments for WT and F/A, $n = 3$ for 4 A; mean ± SD; one-way ANOVA and Tukey's multiple comparison test; ns (no significant difference): $p \geq 0.05$; **$p = 0.0069$; ****$p < 0.0001$; $p$ value between the RAD51 only group and the RAD51 + RPA + CTRB WT group is 0.9999; for the RAD51 + CTRB-F/A group is 0.9992; for the RAD51 + CTRB 4 A group is 0.9737). The samples were derived from the same experiment and the gels were processed in parallel. Source data are provided as a Source Data file.

## Impact of CTRB-F/A and CTRB-4A mutations on HR mediator activity

To determine the contribution of RAD51 interaction and DNA binding by the CTRB to the HR mediator activity of BRCA2, we introduced the F/A and 4 A mutations, singly or in combination, into mini-BRCA2, and then expressed the mutant BRCA2 polypeptides with DSS1 in insect cells. All three mutant mini-BRCA2 species could be purified using the procedure developed for the wild-type counterpart (Supplementary Fig. 3a).

First, we asked how the two CTRB mutations affect RAD51 interaction. Affinity pulldown via the Flag tag on the BRCA2 polypeptides revealed robust interaction of RAD51 with wild-type mini-BRCA2 and mini-BRCA2[4A], but much less RAD51 associated with mini-BRCA2[F/A] mutant (Fig. 3e, lane 9). We note that the large negative impact of the CTRB-F/A mutation on the affinity for RAD51 is consistent with the fact that CTRB associates with oligomeric RAD51, with BRC4 being only capable of interacting with monomeric RAD51[24,48,49] (Fig. 3e). Thus, the residual level of RAD51 interaction observed for mini-BRCA2[F/A] is very likely conferred by the BRC4 repeat in this BRCA2 polypeptide.

We also tested the various mini-BRCA2 species for their DNA binding activity by EMSA. The results revealed that mini-BRCA2-DSS1 possesses a significantly higher affinity for ssDNA than CTRB, but has an affinity for dsDNA similar to that of CTRB (Fig. 3f and Supplementary Fig. 2a). Notably, the 4 A mutation decreases the ability of mini-BRCA2 to bind dsDNA while exerting a lesser effect on ssDNA binding (Fig. 3f and Supplementary Fig. 3b, c). We also verified that the CTRB-F/A mutation has no measurable impact on DNA binding by mini-BRCA2 (Fig. 3f and Supplementary Fig. 3) and that the effect of the CTRB-F/A-4A double mutation on DNA binding by mini-BRCA2 is indistinguishable from that exerted by the 4 A mutation alone (Supplementary Fig. 3c).

## Requirement for CTRB activities in the HR mediator function of BRCA2

Next, we interrogated how the CTRB-F/A and 4 A mutations, singly and in combination, affect the HR mediator activity of mini-BRCA2. Importantly, analysis of the mini-BRCA2 mutants bearing either the CTRB-F/A, 4 A, or both mutations revealed that all three mutant species are compromised for HR mediator activity (Fig. 3g), with the double mutant being much more severely impaired in this regard than either of the single mutants (Fig. 3g). Altogether, we conclude that DNA binding and RAD51 interaction by the CTRB are both important for the HR mediator activity and that the simultaneous loss of both CTRB attributes leads to a more severe impairment of BRCA2 function (Fig. 3g).

## RAD51 targeting ssDNA by CTRB and mini-BRCA2

Previously, we showed that BRCA2 and mini-BRCA2 preferentially targets RAD51 to ssDNA when an excess of duplex DNA is present[10,12]. Here, we asked whether CTRB alone would promote RAD51-loading onto ssDNA with the analysis being conducted alongside mini-BRCA2 known to possess such RAD51 targeting activity[10,12]. In this assay, the influence of CTRB and mini-BRCA2 on the partition of RAD51 between two DNA molecules, one being a biotinylated 80-mer ssDNA-immobilized on magnetic resin and the other dsDNA in solution, was examined (Fig. 4a, i). RAD51 was eluted from the resin by SDS treatment and analyzed by SDS-PAGE to determine the effects of the BRCA2 species in RAD51 partitioning. As shown in Fig. 4a, ii, without CTRB, RAD51 resided in the supernatant (i.e., the fraction that contains the dsDNA trap). Interestingly, the addition of CTRB led to an enrichment of RAD51 in the bead (ssDNA) fraction by 5–10-fold. However, neither the 4 A or F/A CTRB mutant was effective in RAD51 delivery to ssDNA (Fig. 4a, ii). Importantly, a parallel test of mini-BRCA2 revealed a strong RAD51 targeting defect in the F/A and F/A-4A mutants (Fig. 4a, iii). Thus, the CTRB makes a significant contribution to the ssDNA targeting of RAD51 within the context of BRCA2.

## Role of CTRB attributes in DNA damage repair, HR, and RAD51 focus formation

Having established biochemically that both RAD51 interaction and DNA binding by the CTRB are crucial for maximal HR mediator activity, we proceeded to interrogate whether these CTRB attributes are relevant for DNA damage repair in cells. For this, we constructed DLD1 (BRCA2−/−) cell lines stably expressing 2xMBP-full-length BRCA2 species that have the wild-type sequence or harbor the CTRB-F/A, the CTRB-4A, or CTRB-F/A-4A mutation. First, we verified by immunoblot analysis that all four full-length BRCA2 species were expressed at comparable levels and, as determined by sub-cellular fractionation, that they were properly localized in the nucleus (Supplementary Fig. 4a). Next, we treated the various BRCA2 cell lineages with the DNA cross-linking agent MMC or the polyADP-ribose polymerase 1 (PARP) inhibitor Rucaparib, followed by an examination of their clonogenic survival. As expected, cells that harbored the empty vector displayed hypersensitivity to MMC and Rucaparib (Fig. 5a), whereas expression of WT BRCA2 led to a marked enhancement of resistance to both DNA damaging agents[10,33,50,51]. However, cells expressing BRCA2 variants that harbor either the CTRB-F/A or 4 A mutation were clearly sensitized to MMC and Rucaparib, and, importantly, cells that expressed the double BRCA2 mutant exhibited higher sensitivity to these agents than either of the single mutants (Fig. 5a).

We determined the impact of the CTRB mutations on HR proficiency within the context of full-length BRCA2 in two cell models: (1)

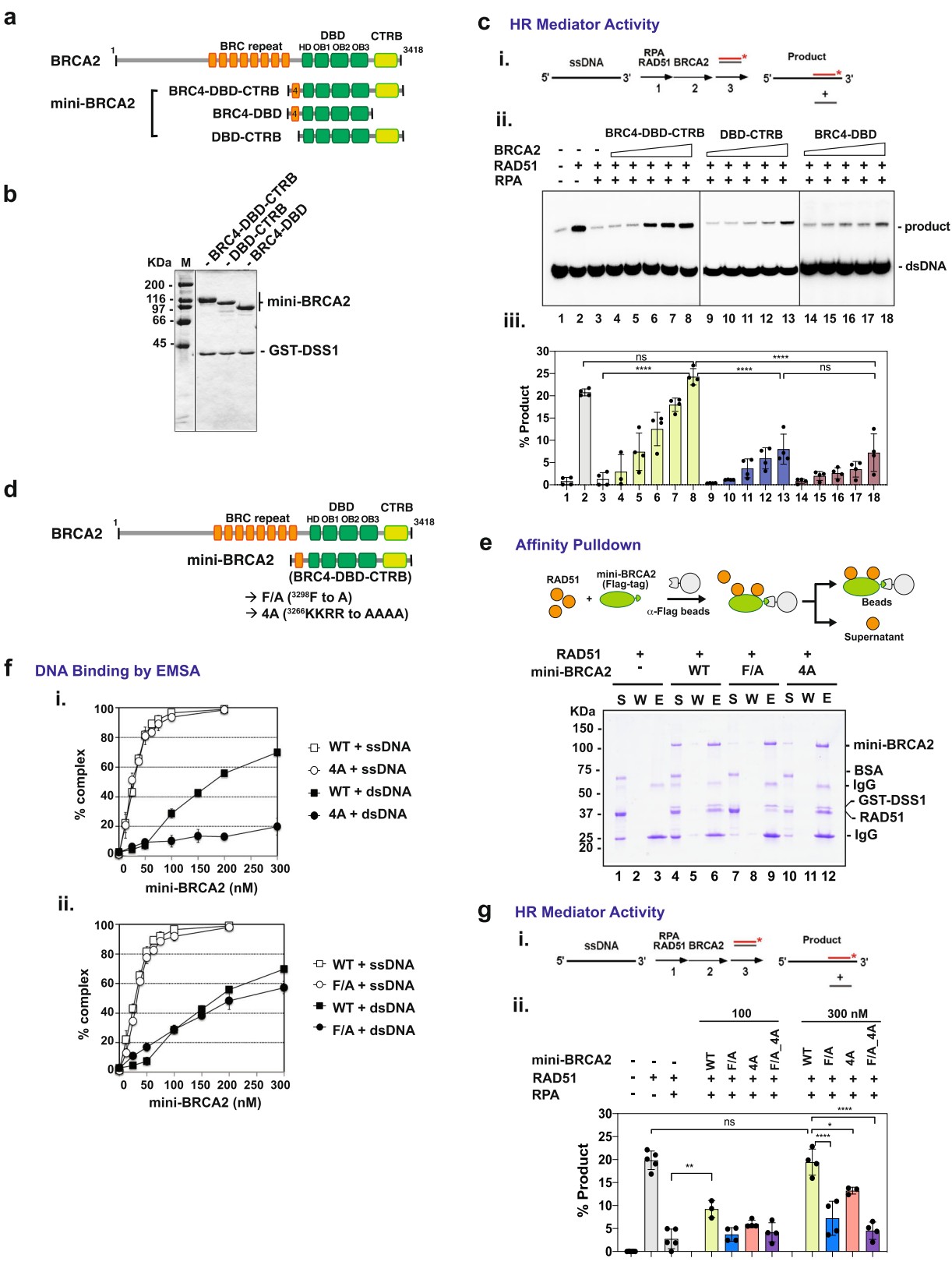

HeLa cells that have an integrated DR-GFP reporter wherein endogenous BRCA2 is depleted with siRNA with either wild type or mutant BRCA2 being ectopically expressed together with I-SceI (Supplementary Fig. 4b)[10,52] and (2) DLD1 cells stably expressing wild type or mutant BRCA2 and ectopically expressing a CRISPR/Cas9-mediated gene targeting system[9,53]. We found in both cell models that neither of the BRCA2 CTRB-F/A and 4 A mutant is as efficacious as wild type

BRCA2 in restoring HR efficiency, while the double CTRB variant is strongly impaired in biological activity (Fig. 5b, c).

Next, we examined the assembly of DNA damage-induced nuclear RAD51 foci as an independent test for BRCA2 HR mediator activity and determined the impact of the CTRB mutations on this activity. Specifically, DLD1 cells stably expressing either wild type BRCA2 or BRCA2 harboring the CTRB-F/A, 4 A, or the F/A-4A double mutation were

**Fig. 3 | Contribution of CTRB functional attributes to HR mediator activity of BRCA2. a** Schematic of BRCA2 species tested. **b** SDS-PAGE of purified BRC4-DBD, DBD-CTRB, and BRC4-DBD-CTRB complexed with GST-DSS1. The experiment was repeated three times with similar results. Source data are provided as a Source Data file. **c** HR mediator activity of BRCA2 species. **(i)** Schematic of the DNA strand exchange assay. The indicated BRCA2 species (60, 120, 180, 240, and 300 nM) were tested for HR mediator activity **(ii)** and the results were quantified and graphed in **(iii)**. ($n = 4$ biologically independent experiments; mean value ± SD; one-way ANOVA and Tukey's multiple comparison test; ns (no significant difference): $p \geq 0.05$; ****$p < 0.0001$. $p$ value between the RAD51 only group and the RAD51 + RPA + mini-BRCA2 (300 nM) group is 0.7872; between DBD-CTRB and BRC4-DBD is 0.9999.) The samples were derived from the same experiment and the gels were processed in parallel. Source data are provided as a Source Data file. **d** Schematic of mini-BRCA2 species that harbor the CTRB-F/A or CTRB-4A mutation. **e** Anti-Flag affinity pulldown assay to test for the interaction of mini-BRCA2/DSS1 (WT, CTRB-F/A, or CTRB-4A, 3 µg each) and RAD51 (8 µg).

S Supernatant containing unbound proteins, W Wash, E SDS-eluate. The experiment was repeated three times with similar results. Source data are provided as a Source Data file. **f** Binding of ssDNA and dsDNA by mini-BRCA2/DSS1. **(i)** WT and CTRB-4A, **(ii)** WT and CTRB-F/A. Note that the same wild-type DNA binding data were plotted in panels (i) and (ii) for comparison with the 4 A mutant (i) or F/A mutant (ii). ($n = 3$ biologically independent experiments; mean value ± SD). The representative gel images are shown in Supplementary Fig. 3b. Source data are provided as a Source Data file. **g** **(i)** Schematic of the DNA strand exchange assay. **(ii)** HR mediator activity of mini-BRCA2/DSS1 (WT, CTRB-F/A, CTRB-4A, or CTRB-F/A-4A at 100 and 300 nM) was graphed. ($n = 5$ (sample number from four biologically independent experiments) for RAD51, RPA + RAD51; $n = 3$ for 100 nM WT, 300 nM 4 A; $n = 4$ for the rest of these groups); mean value ± SD; one-way ANOVA and Tukey's multiple comparison test; ns (no significant difference): $p = 0.9999$; *$p = 0.01219$; **$p = 4.365 \times 10^{-3}$; ****$p < 0.0001$). Representative gel images are shown in Supplementary Fig. 3d. Source data are provided as a Source Data file.

---

exposed to MMC and then scored for RAD51 foci (Fig. 5d and Supplementary Fig. 4c). Again, while the CTRB-F/A and 4 A mutations significantly reduced the number of cells with five nuclear RAD51 foci post-MMC treatment, the double mutation engendered a much stronger defect in RAD51 focus assembly (Fig. 5d). Taken together, results from our cell-based analyses provide compelling evidence for a role of both CTRB attributes, namely, RAD51 interaction and DNA binding, in DNA damage repair and HR.

### Requirement for CTRB attributes in replication fork protection

In addition to its involvement in DSB repair, BRCA2 is also required for the protection of stalled replication forks against nucleolytic attrition by the MRE11 nuclease. This becomes apparent upon exposure of cells to hydroxyurea (HU), which depletes the cellular nucleotide pool needed for DNA synthesis[33,54]. The replication fork protection attribute of BRCA2 is dependent on RAD51, and the CTRB-S3291A mutation impairs this BRCA2 attribute[33].

We sought to investigate the impact of the BRCA2 CTRB-F/A, 4 A, and F/A-4A mutations on replication fork protection. To this end, we used the DNA fiber assay to monitor the tract lengths of nascent DNA in DLD1 BRCA2−/− cells expressing full-length wild type or mutant BRCA2. We labeled cells with the thymidine analogs 5-Chloro-2′-deoxyuridine (CldU) followed by 5-Iodo-2′-deoxyuridine (IdU) before arresting DNA replication for 5 h with 2 mM HU (Fig. 6a). Fork degradation was monitored by determining the ratios of IdU/CldU tract length, as previously described[54]. Consistent with published results[33,55,56], the absence of BRCA2 led to a significantly reduced median IdU/CdU tract length compared to control cells indicative of nascent DNA degradation, whereas DLD1 cells expressing wild-type BRCA2 showed a median IdU/CldU tract length ratio very similar to that of untreated cells (Fig. 6a, iii and Supplementary Fig. 5). Importantly, reduced median IdU/CldU tract length ratios were seen in DLD1 cells expressing any of three BRCA2 CTRB mutants, indicative of a role of the DNA and RAD51 binding attributes of the CTRB in replication fork preservation (Fig. 6a, iv). Fork attrition in mutant cells resulted from MRE11 nuclease activity, as treatment with mirin, an inhibitor of MRE1[33,57], largely prevented nascent DNA strand degradation in uncomplemented cells and cells expressing the BRCA2 mutants (Fig. 6a, iv).

### Discussion

BRCA2 is required for DNA damage repair and replication fork preservation, but there remain major knowledge gaps regarding its mechanisms of action therein. In particular, it has remained unclear how the C-terminus of BRCA2 functions in the aforementioned processes and how it contributes to cell fitness, cancer suppression, and therapy response. This portion of BRCA2 is encoded by gene

exon 27 and was shown, over two decades ago, to harbor a RAD51 interaction domain[28], which we refer to as the CTRB. Later studies by others testing cell lines and mice deleted for the CTRB have yielded additional, compelling evidence for DNA damage repair and replication fork preservation roles for this BRCA2 region. Specifically, mice with a homozygous germline deletion of Brca2 gene exon 27 exhibit a decrease in perinatal viability, decreased survival, and also pronounced tumor susceptibility[30]. Mouse cells that harbor the *lex1* (deleted for part of exon 26 and the entire exon 27) and *lex2* (deleted for the entire exon 27) Brca2 alleles are highly sensitive to γ−radiation and prone to senescence[29]. Importantly, Brca2[lex1/lex2] mutant cells are compromised for HR, as revealed through testing with the DR-GFP HR reporter[31]. Fusing the CTRB to select BRC repeats with and without the DBD generates functional BRCA2 polypeptides that complement PARPi sensitivity, loss of DNA damage-induced RAD51 focus formation, and HR deficiency of BRCA2 mutant cells[34,50]. The CTRB[S3291A] mutation, which partially impairs the RAD51 interaction attribute (Supplementary Fig. 2c), renders cells sensitive to replication stress stemming from an accelerated rate of replication fork attrition[33,58]. Thus, the CTRB clearly fulfils a crucial role in HR, RAD51-mediated DNA damage repair, and the protection of stressed DNA replication forks[31–35].

Herein, we have employed integrated bioinformatic, biophysical, biochemical, and cell-biological approaches to decipher the functions of the CTRB. Specifically, by bioinformatic and NMR analyses, we have provided evidence that CTRB binds both ssDNA and dsDNA. We are able to describe CTRB in possessing a sequential arrangement of MORFs that are likely subject to regulation via post-translational modifications. Importantly, we have provided biochemical evidence that the CTRB is a standalone functional HR mediator module within BRCA2 capable of (i) targeting RAD51 to ssDNA when an excess of duplex DNA is present (Fig. 4a), (ii) stimulating DNA strand exchange efficiency (Fig. 4e), and (iii) allowing RAD51 to utilize an RPA-coated ssDNA template to perform DNA strand exchange (Fig. 2f).

Through the isolation of separation-of-function mutants differentially inactivated for RAD51 interaction (CTRB-F/A) or DNA binding (CTRB-4A), we have shown that both functional attributes are important for the HR mediator activity of the CTRB. Notably, and in congruence with results from a biological examination of functional BRCA2 polypeptides[10,12,34], conjugation of the CTRB to BRC4-DBD leads to a marked enhancement of RAD51-mediated DNA strand exchange when RPA-coated ssDNA is used as template (Fig. 3c). Within the context of BRC4-DBD-CTRB (mini-BRCA2), the introduction of either the CTRB-F/A or CTRB-4A mutation exerts a significant negative impact on the HR mediator activity of this polypeptide, while the simultaneous introduction of both CTRB mutations strongly diminishes mediator activity (Fig. 3g). These

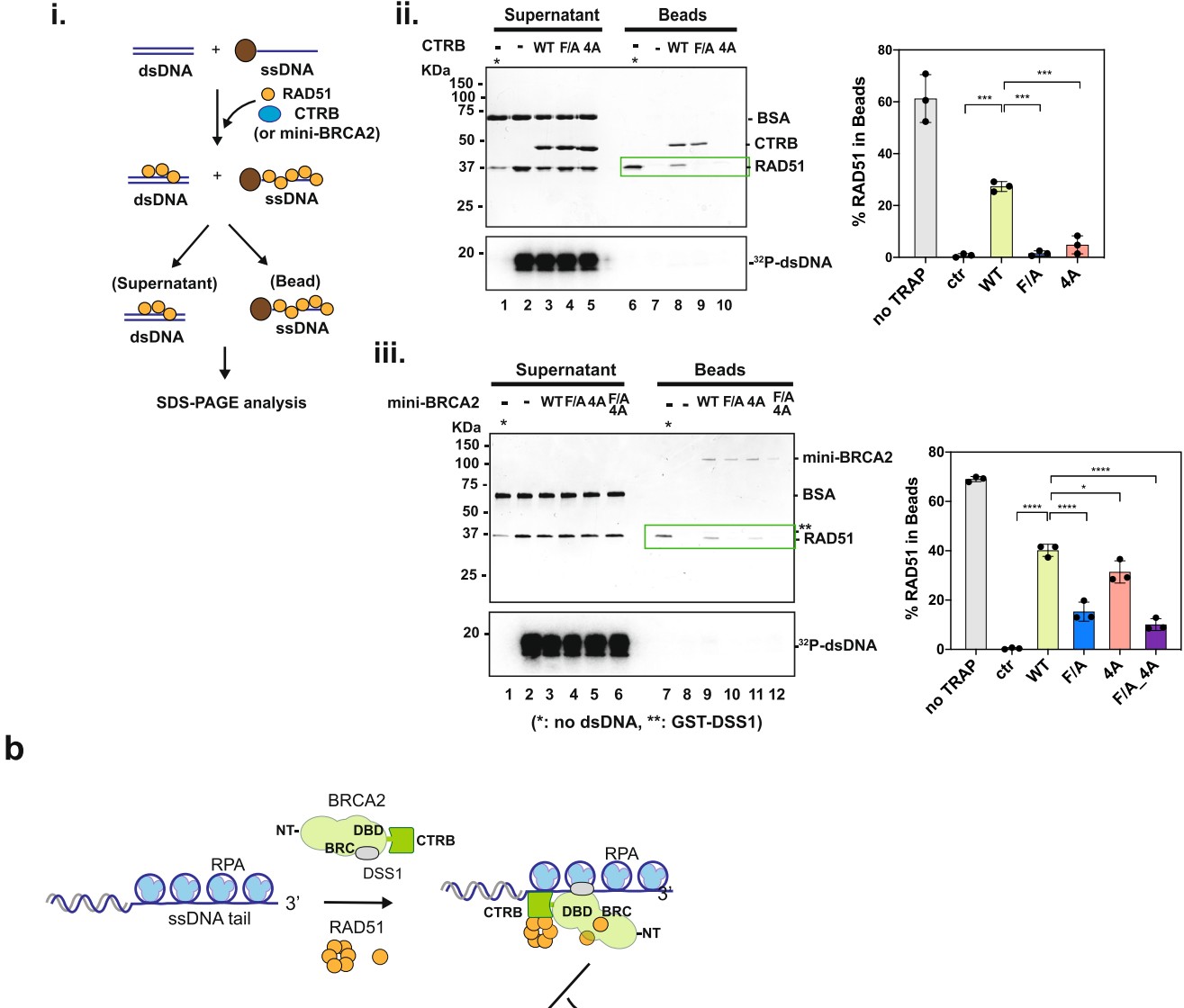

**Fig. 4 | Targeting of RAD51 to ssDNA by CTRB and mini-BRCA2. a** (**i**) Schematic of the assay. The magnetic resin containing ssDNA was incubated with RAD51 and CTRB (**ii**), or mini-BRCA2/DSS1 (**iii**) with and without an excess of radiolabeled dsDNA added as a protein trap. Proteins bound to the ssDNA (Beads) were eluted and analyzed alongside the reaction supernatant(Supernatant) by SDS-PAGE and Coomassie blue staining. The results were quantified and graphed. Phosphorimaging analysis of the dried gel verified that the radiolabeled dsDNA trap remained in the Supernatant fraction. (*, no dsDNA trap; **, GST-DSS1) (n = 3 biologically independent experiments; mean value ± SD; one-way ANOVA and Dunnett's multiple comparison test; ***$p < 0.001$; ****$p < 0.0001$. $p$ value between the CTRB WT group and the control (ctr) group is $1.172 \times 10^{-4}$; for the CTRB-F/A group is $1.512 \times 10^{-4}$; for the CTRB 4 A group is $4.127 \times 10^{-4}$; between the mini-BRCA2 WT group and the 4 A group is 0.01079.) Source data are provided as a Source Data file. **b** Model for CTRB function in RAD51 presynaptic filament assembly. Through its interaction with oligomeric RAD51 and DNA engagement, the CTRB makes a crucial contribution toward the timely assembly of the RAD51 presynaptic filament.

results also provide evidence that BRC repeats, the OB-fold containing DBD, and the CTRB co-operate in imparting maximal HR mediator activity to BRCA2.

Our results also reveal the functional interplay between CTRB and the BRC-DBD domain. Specifically, our biochemical data show clearly that combining both entities, as in the case of mini-BRCA2, leads to a significant enhancement of HR mediator activity. Thus, CTRB and BRC-DBD are not simply redundant functional HR mediator modules in BRCA2, but, rather, they act in concert to enhance RAD51 presynaptic filament assembly. We note that our results are incongruence with

published results showing that the CTRB is important for the biological efficacy of BRCA2-derived polypeptides[34].

In addition to our biochemical and biophysical studies, we have also carried out extensive biological analyses of full-length BRCA2 variants harboring either the CTRB-F/A, CTRB-4A, or CTRB-F/A-4A mutation in order to elucidate the biological roles of the CTRB-RAD51 interaction and DNA binding attributes. Altogether, the results from these endeavors reveal that both attributes are indispensable for (i) HR proficiency as measured by two distinct, well-established reporters; (ii) conferring cellular resistance to DNA damaging agents; (iii) the

**Fig. 5 | Significance of CTRB attributes in DNA damage repair. a** Clonogenic survival of DLD1 cells expressing wild-type BRCA2 or BRCA2 variants with the indicated CTRB mutation (F/A, 4 A, or F/A-4A) after treatment of MMC or Rucaparib. EV: empty vector. ($n = 4$ biologically independent experiments for MMC, mean value ± SEM; $n = 3$ biologically independent experiments for Rucaparib, mean value ± SD). Source data are provided in the Source Data file. **b** HR proficiency was measured using the DR-GFP gene conversion reporter in HeLa cells depleted for endogenous BRCA2 and expressing BRCA2 or BRCA2 variants with the indicated CTRB mutation. Ctr (control): not treated. ($n = 5$ biologically independent experiments; mean value ± SD; unpaired $t$-test (two-tailed); *$p = 4.928 \times 10^{-2}$; **$p < 0.01$; ***$p < 0.001$; ****$p < 0.0001$. $p$ value between the WT group and the EV group is $1.608 \times 10^{-4}$; for the F/A group is $1.116 \times 10^{-3}$; between the F/A group and the F/A_4 A group is $2.898 \times 10^{-3}$; between the 4 A group and the FA_4 A group is $7.481 \times 10^{-4}$).

Source data are provided as a Source Data file. **c** HR proficiency was measured in DLD1 cells expressing BRCA2 and BRCA2 variants with the indicated CTRB mutation using a CRISPR/Cas9-based gene targeting assay. Ctr (control): not treated. ($n = 3$ biologically independent experiments; mean value ± SEM; one-way ANOVA and Tukey's multiple comparison test; **$p = 1.625 \times 10^{-3}$; ****$p < 0.0001$. $p$ value between the WT group and the 4 A group is $1.183 \times 10^{-5}$.) Source data are provided as a Source Data file. **d** RAD51 focus formation after MMC treatment. Representative micrographs of RAD51 foci from Supplementary Fig. 4c are shown. ($n = 7$ biologically independent experiments for WT, $n = 4$ for EV and 4 A, $n = 3$ for F/A and F/A_4 A; mean value ± SEM; one-way ANOVA and Tukey's multiple comparison test; ns (no significant difference): $p = 0.2212$; ****$p < 0.0001$. $p$ value between the WT group and the F/A group is $7.100 \times 10^{-5}$; between the 4A group and the F/A_4A group is $8.623 \times 10^{-5}$.) Source data are provided as a Source Data file.

competence of cells to assemble DNA damage-induced RAD51 foci; and (iv) the protection of stressed DNA replication forks against MRE11-dependent nucleolytic attack. As we have observed in the biochemical experiments, the CTRB-F/A-4A mutant is significantly more impaired in biological activity than either the CTRB-F/A or CTRB-4A mutant.

Based on our in vitro data and findings in cells, we propose that the CTRB fulfills distinct mechanistic and biological roles in HR and

DNA replication fork preservation. At resected DSBs, the DNA binding activity of the CTRB assists in the delivery of bound oligomeric RAD51 to RPA-coated ssDNA to seed the assembly of the presynaptic filament to mediate DNA strand invasion and strand exchange (Fig. 4b) crucial for the completion of HR-mediated DSB repair. However, at stressed DNA replication forks, the role of CTRB is to engage regressed replication forks, generated by one of several ATP-dependent DNA motor

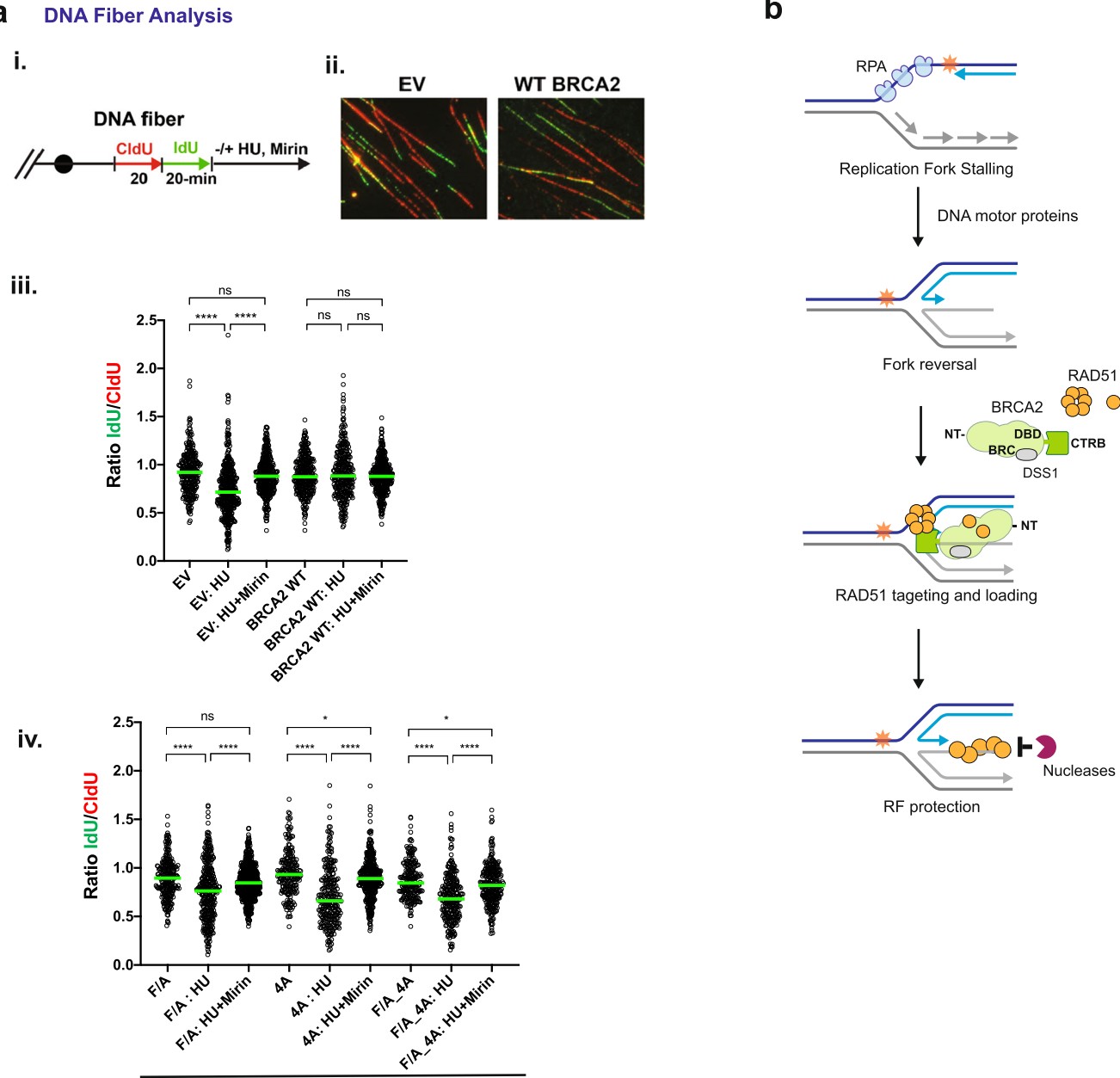

**Fig. 6 | Significance of CTRB attributes in replication fork protection.**
**a** Schematic of DNA fiber analysis (**i**) and representative micrographs (**ii**). Dot plots of IdU to CldU tract length ratios for individual replication forks in unperturbed, and HU-treated cells with or without mirin (**iii**, **iv**). The median value of 100–150 CldU and IdU tracts from three and four independent experiments for unperturbed and treated cells, respectively, is indicated. Kruskal–Wallis test followed by Dunn's multiple comparisons test. (ns (no significant difference): $p \geq 0.05$; *$p < 0.05$; ****$p < 0.0001$. $p$ value between the EV group and the EV: HU + Mirin group is

0.3046; between the F/A group and the F/A: HU + Mirin group is 0.09772; between the 4A group and the 4A:HU + Mirin group is $4.396 \times 10^{-2}$; between the F/A_4A group and F/A_4A:HU + Mirin group is 0.04908.) Representative micrographs of DNA fibers (EV: empty vector. WT BRCA2 cells) from Supplementary Fig. 5 are shown. Source data are provided as a Source Data file. **b** Model for CTRB function at regressed replication forks. Through its interaction with oligomeric RAD51 and DNA engagement, the CTRB helps seed the assembly of a RAD51 protein filament of the arm of a regressed replication fork that harbors the free DNA end.

proteins, via its DNA binding attribute and loading the associated RAD51 protomers onto the regressed forks to protect them against nucleolytic digestion catalyzed by MRE11 and possibly other nucleases (Fig. 6b). We wish to highlight the finding that the CTRB-4A DNA binding mutant is somewhat less severely impaired than the CTRB-F/A mutant in assays that measured HR efficiency and DNA damage-induced RAD51 focus formation (Fig. 5b–d), but is just impaired as the latter mutant in the protection of stressed DNA replication forks against nucleolytic attrition (Fig. 6a). Based on these results, we surmise that the ability of CTRB to bind both ssDNA and dsDNA (Fig. 2c

and Supplementary Fig. 2b) allows it to engage the ss-dsDNA junction of regressed replication forks to facilitate RAD51 nucleoprotein filament assembly (Fig. 6b). We note that our studies and the experimental systems described herein will be highly valuable for addressing how cancer-associated mutations in the DBD and CTRB affect the functions of BRCA2 in DNA damage repair and DNA replication fork protection.

Our work also sheds light on how higher organisms adapt to stress within short evolutionary periods via the usage of disorder and the relatively rapid formation of compact but highly controllable modules,

as we have shown for the BRCA2 CTRB. Such function is often disguised by low levels of conservation, and therefore we hope to create an awareness of how pliable these modules are for spontaneous mutations on significantly shorter timescales. In addition, BRCA2 is a key tumor suppressor, with a plethora of mutations found in various types of cancers, with thousands of genetic variants of unknown significance (VUS) in the gene[41,59,60]. Notably, the C-terminal region is a hub for cancer VUSs which cannot be rationally explored in the absence of functional understanding. We now provide an extensive molecular exploration of this region, pointing directly to key regions for functional genetic and clinical genetic prioritization. Moreover, we elucidate the marked functional importance of motifs in CTRB for DNA repair and genome maintenance, which further will open new possibilities for clinical genetic advances. Our experimental systems should be valuable for testing the functional and clinical impact of BRCA2 CTRB variants of unknown significance (VUS), e.g. K3267N and R3269G that alter Cluster A DNA binding residues and F3298I that could affect RAD51 interaction[61].

## Methods

### Plasmids
The cDNA that encodes C-terminally Flag-tagged CTRB (BRCA2 amino acid residues 3194–3418) was introduced into the NcoI/XhoI sites of the pET32a vector for protein expression of in *E. coli*. CTRB mutations, namely F3298 to A (designated CTRB-F/A), KKRR(3266–3269) to AAAA (designated CTRB-4A), or the F/A-4A double mutation were generated by site-directed mutagenesis in pET32a-CTRB or pDEST8-mini-BRCA2 harboring BRC4 (BRCA2 amino acid residues 1496–1596) conjugated to the DBD and CTRB (BRCA2 amino acid residues 2477–3418) with C-terminal His6-Flag double tag[10]. We also constructed pFastbac-BRC4-DBD-CTRB-His6, pFastbac-BRC4-DBD-His6, and pFastbac-DBD-CTRB-His6. For cellular studies, we introduced the CTRB-F/A, 4 A, and the F/A-4A mutations into phCMV-2xMBP-BRCA2[11]. Construction of other protein expression plasmids, namely, pDEST20-DSS1, pET11-RAD51, and pET11d-RPA, has been described in refs. [10,62]. Variants of pGEX-TR2 (BRCA2 amino acid residues 3265–3330) were generated by site-directed mutagenesis. The aforementioned plasmids were sequenced to verify that they contain no unwanted mutation.

### Cell culture
HeLa cells carrying a single copy of the DR-GFP reporter (HeLa-DR13-9 cells)[52] were cultured in DMEM (Gibco, Catalog #11965-092) and DLD1 cells (BRCA2−/−, Horizon Discovery) were maintained in RPMI (Gibco, Catalog # 11875-093). DLD1 cells were transfected with 2 μg phCMV-2xMBP-BRCA2 (encoding full-length BRCA2) or the indicated mutant variant using Lipofectamine 2000 (Invitrogen) per manufacturer's instructions, and single-cell colonies stably expressing wild type or mutant BRCA2 were isolated and maintained in 0.5 mg/ml G418.

### Protein purification
CTRB and mutants: *E. coli* Rosetta cells (Novagen) harboring pET32-CTRB were grown in 2x Luria broth (LB) media supplemented with 33 μg/ml chloramphenicol and 100 μg/ml ampicillin at 37 °C. When the $OD_{400}$ reached 0.8, cells were treated with 1 mM IPTG for 4 h. Cells were harvested and stored at −80 °C. All purification steps were carried out at 0–4 °C. A 10 g cell pellet was resuspended in buffer K (20 mM $KH_2PO_4$, pH 7.5, 10% glycerol, 0.5 mM EDTA, 1 mM DTT, 0.01% Igepal CA-630) containing a cocktail of protease inhibitors (PIs) (1 mM PMSF and 2 μg/ml each of aprotinin, chymostatin, leupeptin, and pepstatin A), and 150 mM KCl. Cell extract was prepared by sonication and clarified by ultracentrifugation at 100,000 × g for 90 min, before being incubated with 6 ml Ni-NTA agarose (Qiagen) and 10 mM imidazole for 1 h on a rotary mixer. The resin was collected in a chromatography column and washed with 20 ml each of buffer K containing 150, 1000,

and 50 mM KCl. Bound proteins were eluted with 300 mM imidazole in buffer K containing 50 mM KCl. Peak CTRB fractions were pooled and fractionated in a Source S column (8 ml matrix) and eluted with a 150 ml gradient of 30 to 500 mM KCl in buffer K. Factions containing CTRB were pooled and mixed with 1 ml of anti-Flag M2 affinity gel (Sigma) for 2 h. After washing with buffer K containing 100 mM KCl, proteins were eluted with 5 ml of 100 μg/ml Flag peptide in buffer K containing 100 mM KCl. The elution fractions were diluted twofold with buffer K and applied onto a 1 ml Mono-Q column and bound proteins were eluted with a 15 ml gradient of 30 to 500 mM KCl in buffer K. Fractions containing CTRB were pooled and concentrated to 5 mg/ml in an Amicon Ultra centrifugal device (EMD Millipore). The concentrated protein (-300 μg total) was stored in 2 μl aliquots at −80 °C.

Mini-BRCA2 and mutants complexed with DSS1: Recombinant bacmids encoding mini-BRCA2-His6 or GST-DSS1 were produced in DH10Bac (Thermo Fisher Scientific, Cat # 10361012). Baculoviruses were amplified in Sf9 insect cells (Thermo Fisher Scientific, Cat #B825-01) and complexes of recombinant mini-BRCA2-His6 and GST-DSS1 were produced in High-Five insect cells (Thermo Fisher Scientific Cat #B855-02) co-transfected with the baculoviruses for 48 h at 27 °C. Cells were harvested by centrifugation and stored at −80 °C. Cell extract was prepared by sonication in 30 ml buffer T (25 mM Tris-HCl, pH 7.5, 10% glycerol, 0.5 mM EDTA, 1 mM DTT, 0.01% Igepal CA-630) containing PIs and 300 mM KCl and clarified by ultracentrifugation at 100,000 × g for 90 min. The supernatant was supplemented with 10 mM imidazole and incubated with 6 ml of Ni-NTA agarose for 1 h on a rotary mixer. The resin was washed with 20 ml each of buffer T containing 300 mM, 1 M, and 150 mM KCl. Bound proteins were eluted with 200 mM imidazole in buffer T containing 150 mM KCl. Fractions containing proteins were pooled and mixed with 5 ml of glutathione-Sepharose 4 Fast-flow resin (GE Healthcare) for 1 h on a rotary mixer. The beads were washed with 50 ml each of buffer T containing 8 mM $MgCl_2$, 1 mM ATP and 1 M KCl, and buffer T containing 100 mM KCl. Pooled protein fractions were diluted with an equal volume of buffer T and loaded on a 1 ml Mono-Q (GE Healthcare). The bound proteins were fractionated with a 20 ml gradient of 200 mM to 400 mM KCl in buffer T. Peak fractions containing mini-BRCA2-His6 and GST-DSS1 were pooled and concentrated in an Amicon Ultra centrifugal device (EMD Millipore). The concentrated proteins were stored in small aliquots at −80 °C.

$^{13}C$ and $^{15}N$-His-tagged CTRB fragment (amino acid residues 3,260 to 3,337) was expressed in *E. coli* Rosetta strain transformed pDEST17-HBS plasmid, cultured in M9 minimal medium enriched with $^{13}C$-enriched glucose and $^{15}N$-enriched $NH_4Cl$ and supplemented with 33 μg/ml chloramphenicol, 100 μg/ml ampicillin, at 37 °C. Expression was induced at an $OD_{400}$ of 0.6 with 0.5 mM IPTG and cells were harvested after 2 h of incubation by centrifugation. The collected cells were resuspended in 25 mM Tris-HCl, pH 8.0, 500 mM NaCl, 20 mM imidazole, 1 mM DTT, with protease inhibitors (1 mM PMSF and complete EDTA-free cocktail), and cell lysate was prepared by sonication with Bioruptor® ultrasonicator for 10 min with the 30 s/30 s on and off intervals. Cell debris were removed by centrifugation for 30 min at 4 °C at 17,000 rpm. The clarified lysate was mixed with a 1 ml $Ni^{2+}$-NTA Sepharose resin for 30 min at 4 °C. After washing the resin with 120 ml of the cell breakage buffer, His-BRCA2 CTRB fragment was eluted with 5 ml 25 mM Tris-HCl, pH 8.0, 500 mM NaCl, 500 mM imidazole, and 1 mM DTT. Peak fractions that contained the CTRB fragment were pooled and dialyzed against 1 L PBS, 10 mM DTT, pH 6.0. After dialysis, the protein concentration was measured by Bradford assay and the protein was aliquoted and placed for storage at 4 °C until further use.

BRCA2-TR2 polypeptide: TR2 harboring BRCA2 amino acid residues 3265–3330 or mutant variants tagged N-terminally with GST were expressed in *E. coli* Rosetta cells. For protein purification, clarified bacterial cell lysate (prepared by sonication in 20 mM $KH_2PO_4$, pH 7.5, 300 mM KCl, 10% glycerol, 0.5 mM EDTA, 0.01% Igepal CA-630, and

PIs) was prepared as described above and subject to affinity chromatography on glutathione-Sepharose followed by fractionation in an SP Sepharose column.

RAD51 and RPA: RAD51 was prepared following our published procedure[62] with minor modifications. RAD51 was expressed in *E. coli* BLR pRARE (EMD Millipore) cells using pET11-RAD51 and a 16 h incubation with 1 mM IPTG at 16 °C. The cell lysate was prepared by sonication of resuspended cells (40 g) in 200 ml Cell breakage buffer (50 mM Tris-HCl, pH 7.5, 5 mM EDTA, 1 M mM KCl, 1 mM DTT, 10% sucrose, 0.01% Igepal CA-630 (Sigma), PIs, as described above,) followed by ultracentrifugation (100,000 × *g*, 90 min). The clarified cell lysate was precipitated with 40% ammonium sulfate for 1 h and the precipitated proteins were collected by centrifugation (18,000×*g*, 20 min). The pellets were dissolved in 200 ml buffer K and RAD51 was sequentially fractionated on a Fast-flow Q Sepharose column (GE Healthcare) (40 ml matrix) with a 400 ml gradient of 150 to 600 mM KCl in buffer K, a Macro hydroxyapatite column (Bio-Rad) (8 ml matrix) with a 120 ml gradient of 0 to 300 mM $KH_2PO_4$ in buffer T, and a 1 ml Mono-Q column (GE Healthcare) with a 30 ml gradient of 200 to 500 mM KCl in buffer T. Peak fractions containing RAD51 were pooled, concentrated in an Amicon Ultra-15 centrifugal device (EMD Millipore), aliquoted, and stored at −80 °C.

RPA was prepared following our published procedure[62]. RPA was expressed in *E. coli* Rosetta cells using pET11-RPA and a 16 h incubation with 0.1 mM IPTG at 16 °C. The cell lysate was prepared by sonication of resuspended cells (33 g) in 160 ml buffer T containing 100 mM KCl and PIs. RPA was fractionated sequentially on an Affi-gel blue column (Bio-Rad) (20 ml matrix) with a 140 ml gradient of 0.5 to 2.5 M NaSCN gradient in buffer T, a Macro hydroxyapatite column (Bio-Rad) (8 ml matrix) with a 120 ml gradient of 10 to 140 mM $KH_2PO_4$ in buffer T, and a 1 ml Mono-Q column (GE Healthcare) with a 30 ml gradient of 40 to 300 mM KCl in buffer T. Peak fractions containing RPA were pooled, concentrated in an Amicon Ultra-15 centrifugal device (EMD Millipore), aliquoted, and stored at −80 °C.

### DNA substrates
DNA oligonucleotides (listed in Supplementary Table 2) were purchased from IDT and purified by polyacrylamide gel electrophoresis under denaturing conditions. Where indicated, DNA was labeled with T4 polynucleotide kinase and [γ-$^{32}$P] ATP[47]. Cy-5 labeled oligomers were purchased from IDT. The dsDNA substrates were prepared by annealing equimolar amounts of complementary oligos as described in ref. [47].

### DNA binding assay
The indicated concentration of BRCA2-derived polypeptides was incubated with 10 nM ssDNA or dsDNA in 10 μl of buffer B (35 mM Tris, pH 7.5, 1 mM $MgCl_2$, 1 mM DTT, 100 μg/ml BSA, and 50 mM KCl) at 25 °C for 5 min. Reactions were resolved by native gel electrophoresis in 8% polyacrylamide gels and TAE buffer (50 mM Tris-acetate, pH 7.5, and 0.5 mM EDTA) at 50 mA for 1 h. For detection of $^{32}$P-labeled DNA, gels were dried on a sheet of DEAE paper or Hybond-N (Amersham) and analyzed in Personal Imager FX (Bio-Rad) using ImageLab 5.2.1(Bio-Rad) or Amersham Typhoon phosphorimager (Cytiva) using the ImageQuant 8.2 software (Cytiva). Gels containing Cy-5 labeled DNA were analyzed using a ChemiDoc imaging system and the ImageLab software 5.2. (Bio-Rad). Mean values and standard deviations were calculated and presented using GraphPad Prism 8.4 or Microsoft Excel 16.

### Homologous DNA pairing assay
In the standard reaction (12.5 μl), RAD51 (2 μM) was incubated with ssDNA (150-mer Oligo A; 40 nM) in buffer B containing 1 mM ATP at 37 °C for 5 min. The reaction was initiated by adding homologous dsDNA (consisting of Oligo B hybridized to 5′ $^{32}$P-labeled Oligo C;

40 nM) and 4 mM spermidine hydrochloride.) After 30 min of incubation, the reaction was stopped by incubation with 1 μl of 1% SDS and 1 μl of 10 mg/ml proteinase K for 10 min at 37 °C. The samples were resolved in 8% native polyacrylamide gels in TAE buffer at 4 °C. Gels were dried and imaged and analyzed as described above. To examine HR mediator activity, the ssDNA was first incubated with RPA (500 nM) for 5 min before the addition of RAD51 (2 μM) and the indicated BRCA2 species. Incubation and analysis of reaction mixtures were carried out as described above[47]. Mean values were calculated from the data obtained from three to five independent experiments. Data were processed and plotted using GraphPad Prism 8.4. Standard deviations were calculated and presented as error bars together with the mean values. *p* values were calculated using a one-way ANOVA test and Tukey's multiple comparison test.

### ssDNA targeting assay
In the standard reaction (20 μl), RAD51 (2.7 μM) and CTRB (1.25 μM), or mini-BRCA2 (0.3 μM) were pre-incubated in 35 mM Tris-HCl, pH 7.5. 1 mM $MgCl_2$, 1 mM ATP, 30 mM KCl, 50 ng/μl BSA at 25 °C for 10 min before being incubated with 200 nM $^{32}$P-labeled dsDNA (83-bp, Supplementary Table 2) and 50 nM ssDNA (Biotin-dT80)-immobilized on 4 μl Dynabeads M270 Streptavidin (Invitrogen) at 37 °C for 10 min. Then, the resin was captured with a Magnetic Particle Separator (Roche Applied Science) and washed twice with 20 μl of 35 mM Tris-HCl, pH 7.5. 1 mM $MgCl_2$, 0.1 mM ATP, 30 mM KCl. Proteins were eluted from the resin using 25 μl of SDS-PAGE loading buffer at 37 °C. Half each of the eluate and the supernatant containing unbound proteins were analyzed by SDS-PAGE and Coomassie staining. Gels were dried on a sheet of Hybond-N and subjected to phosphorimaging analysis as described above. The mean values of RAD51 in the resin were calculated from the data obtained from three independent experiments. Data were processed and plotted using GraphPad Prism 8.4. Standard deviations were calculated and presented as error bars together with the mean values. *p* values were calculated using one-way ANOVA and Dunnett's multiple comparison test.

### Affinity pulldown assay
Reactions containing the indicated proteins were incubated in 25 μl buffer T (25 mM Tris-HCl, pH 7.5, 0.5 mM EDTA, 10% glycerol, 0.01% Igepal CA-630) with 150 mM KCl supplemented with 6 U/μl Benzonase on ice for 30 min before being mixed with 15 μl of S-Protein agarose resin (EMD Millipore) for CTRB-RAD51 pull-down, anti-Flag agarose resin (Sigma) for mini-BRCA2 pull-down experiment) for 1 h at 4 °C. The resin was collected by centrifugation and after it was washed three times with 100 μl of buffer T-150 mM KCl, bound proteins were eluted with 25 μl of SDS-PAGE loading buffer at 37 °C. Unbound (Supernatant, S), last wash (W), and elution (E) fractions were analyzed by SDS-PAGE and Coomassie Blue staining. The pulldown experiment shown in Supplementary Fig. 2d was carried out with RAD51 and the indicated GST-tagged TR2 polypeptides in 10 μl buffer K (20 mM $KH_2PO_4$, pH 7.5, 0.5 mM EDTA, 10% glycerol, 0.01% Igepal CA-630) with 150 mM KCl. Protein complexes were captured with 10 μl of glutathione-Sepharose (Cytiva) and analyzed as described above. Stained gel images were scanned and processed with an Epson Perfection V700 photo scanner and ScanSmart or Bio-Rad ChemoDoc and ImageLab 5.2.

### Cell-based HR assay
DR-GFP reporter assay: HeLa cells (1 × 10$^5$) with an integrated DR-GFP reporter[52] were seeded in six-well plates and incubated for 24 h. Then, cells were transfected with 2.5 μl RNAimax (Invitrogen) and 50 ng siRNA (si-BRCA2 (UUGGAGGAAUAUCGUAGGUAAUU (Dharmacon))[10] or control siRNA (Sigma, Cat. # SIC001). Twenty-four hours after the first transfection, cells were transfected with 3.5 μl Lipofectamine 2000 (Invitrogen), 2 μg of phCMV-2xMBP-BRCA2 (wild type or the indicated CTRB mutant variant), and 0.6 μg of I-SceI plasmid

(pCBASce), as described[10]. Following a 72 h incubation, cells were collected in 300 µl PBS with 5% FBS and GFP⁺ cells were quantified by flow cytometry with an LSRII instrument (BD Bioscience) and FlowJo'M vl0 software (BD Biosciences). Mean values of GFP + cells were calculated from the data obtained from five independent experiments. Data were processed and plotted using GraphPad Prism 8.4. Standard deviations were calculated and presented as error bars together with the mean values. $p$ values were calculated using an unpaired Student's $t$-test.

## CRISPR/Cas9-induced gene targeting assay

DLD1 cells ($4 \times 10^5$) expressing 2xMBP-BRCA2 (wild type or the indicated CTRB mutant) were seeded in each well of six-well plates and incubated for 24 h. Cells were transfected with 1 µg pX330-LMNA1 and 1 µg pCR2.1-CloverLamin, as described in ref. [10], and 4 µl Lipofectamine 2000 (Invitrogen). Following a 72 h incubation, cells were collected in 300 µl PBS with 5% FBS and GFP⁺ cells were quantified by flow cytometry. The mean values of GFP+ cells were calculated from the data obtained from three independent experiments. Standard deviations were calculated and presented as error bars together with the mean values. $p$ values were calculated using a one-way ANOVA test. Protein levels in the samples were examined by SDS-PAGE and Western blotting with anti-BRCA2 (1: 500, Millipore Sigma, Cat. # OP95), anti-Tubulin (1: 3000, Cell signaling Technology, Cat. # 2125), anti-MBP- (1: 1000, Novus Biologicals, Cat. # NB100-66609H), anti-RAD51 (1: 500, Abnova, Cat. #: H00005888-B01P), and anti-Histone H3 (1: 1000, Cell Signaling Technology, Cat. #5192), anti-mouse-IgG-secondary antibody (1:4000, Pierce Cat. ##31450), anti-rabbit-IgG-HRP (secondary antibody, 1:4000 (Sigma #A6154), and anti-mouse-IgG-HRP (secondary antibody, 1:4000, Pierce Cat. ##31450) antibodies.

## Clonogenic survival assay

DLD1 cells expressing 2xMBP-BRCA2 (wild type or the indicated CTRB mutant) were seeded at 300 cells per well in six-well plates. After 24 h incubation, cells were treated with the indicated concentrations of MMC (Fisher Scientific, Cat. #50-448-826) or Rucaparib (Selleck Chemical, Cat. #AG-014699) for 10 days. Cells were fixed with methanol and stained with 0.5% crystal violet in 20% methanol before colonies were counted. Clonogenic survival was calculated from the number of colonies that survived drug treatment as compared to the untreated control. Mean survival values were calculated from the data obtained from four (MMC) or three (Rucaparib) independent experiments. Data were processed and plotted using GraphPad Prism 8.4. Standard errors (SEM) were calculated and presented as error bars together with the mean values.

## Cell fractionation

DLD1 cells expressing 2xMBP-BRCA2 (wild type or CTRB mutant) were grown on a 10-cm plate and collected at -90% confluency. Harvested cells were washed in 1x PBS, and resuspended in 1 ml extraction buffer (10 mM HEPES pH 7.9, 10 mM KCl, 1.5 mM MgCl₂, 0.34 M sucrose, 10% glycerol, 0.075% Igepal CA-630, 4PIs), and incubated on ice for 10 min with occasional vortexing. Nuclei were collected by centrifugation at $6500 \times g$ for 5 min at 4 °C. The collected nuclei were washed for 1 min on ice with extraction buffer without 0.075% Igepal CA-630 and with 1 ml NS-0 buffer (20 mM Hepes pH 7.9, 1.5 mM MgCl₂, 3 mM EDTA, 0.2 mM EGTA, and PIs before being resuspended in 300 µl NS-420 buffer (20 mM Hepes pH 7.9, 1.5 mM MgCl₂, 420 mM NaCl, 0.2 mM EDTA, 0.5 mM DTT, 25% glycerol, 0.5% Igepal CA-630, and PIs), incubated on ice for 20 min and spun at $15,000 \times g$ for 20 min at 4 °C. The nuclear fraction (supernatant) was collected. The cytoplasmic and nuclear fractions were analyzed by SDS-PAGE and Western blotting with anti-MBP (1: 1000, Novus Biologicals Cat. # NB100-66609H), anti-Tubulin (1: 3000, Cell signaling Technology, Cat. # 2125), and anti-Histone H3 (1:1000, Cell Signaling Technology, Cat. #5192), anti-rabbit-IgG-HRP (secondary antibody, 1:4000 (Sigma #A6154). The blots were analyzed with Bio-Rad ChemoDoc and ImageLab v5.2.

## DNA fiber assay

DNA replication tracts were assessed by the single-molecule DNA fiber assay as described previously[54,63]. Briefly, exponentially growing cells were pulse-labeled with 25 µM CldU (20 min), washed three times in warm PBS, and exposed to 250 µM IdU (20 min) in a regular growth medium. Then cells were washed with warm PBS and incubated in a regular growth medium with or without 2 mM HU (Sigma-Aldrich), or HU + mirin (50 µM, Sigma-Aldrich). Labeled cells were collected by scraping, resuspended in ice-cold PBS at $4 \times 10^5$ cells/ml, and 2 µl of this suspension were spotted onto a glass slide and lysed with 7 µl spreading buffer (0.5% SDS, 200 mM Tris-HCl, pH 7.4, 50 mM EDTA). After a 5-min incubation, the slides were tilted, allowing the DNA to spread. Slides were air-dried and fixed in methanol:acetic acid (3:1), rehydrated in PBS for 10 min and denatured in 2.5 M HCl for 1 h at room temperature. Slides were then rinsed in PBS and blocked in PBS + 0.1% Triton X-100 (PBS-T) + 5% BSA for 1 h at room temperature. Rat anti-BrdU (1:100, Bio-Rad, Cat #MCA2060T) and mouse anti-IdU (1:100, Becton Dickinson, Cat #347580) antibodies were then applied in blocking solution to detect CldU and IdU, respectively. After a 1 h incubation, slides were washed in PBS and stained with Alexa Fluor 488-labeled goat anti-mouse IgG (1:300; Thermo Fisher Scientific, Cat #A-11029) and Alexa Fluor-594-labeled goat anti-rat antibody (1:300; Thermo Fisher Scientific, Cat #A-11007). Slides were mounted in Prolong Gold Antifade (Thermo Fisher Scientific) and held at 4 °C until image acquisition. Replication tracts were imaged on a Zeiss Axio-Imager.Z2 microscope equipped with ZEN Blue software (Carl Zeiss Microscopy) using a 63x oil objective. DNA tracts were measured using ImageJ v1.53 software. Data are from three to four independent experiments with 100–150 DNA fibers each. Data were plotted in GraphPad Prism 8.4. Statistical analysis was conducted using the Kruskal–Wallis test followed by Dunn's multiple comparisons test.

## RAD51 focus formation

Four-well chamber slides were seeded with 70,000 cells per chamber 48 h before treatment with 1 µM MMC for 16 h. Cells were washed once in PBS then pre-extracted with CSK buffer (10 mM PIPES, pH 7.0, 100 mM NaCl, 300 mM sucrose, and 3 mM MgCl₂) containing 0.2% Triton X-100 for 2 min at 25 °C and then washed with PBS once before fixation with 2% paraformaldehyde for 15 min. Cells were permeabilized in PBS + 0.25% Triton X-100 for 15 min at 25 °C, washed with PBS, and blocked with PBS + 0.1% Tween-20 (PBS-T) containing 1% bovine serum albumin (BSA) for 2 h at 25 °C. Chamber slides were incubated with primary antibody (α-RAD51 (H-92; Santa Cruz Biotechnology, Cat. # sc-8349; 1:1000)) in PBS-T + 1% BSA overnight and then washed with PBS-T and incubated with Alexa Fluor-594 goat anti-rabbit secondary antibody (Thermo Fisher Scientific; 1:750, Cat #A-11037) in PBS-T + 1% BSA for 45 min at 25 °C. After several washes in PBS, chamber slides were mounted in ProLong Gold with DAPI (Thermo Fisher Scientific). Images were taken using a 63× oil objective and a Zeiss Axio-Imager.Z2 microscope equipped with Zen Blue software (Carl Zeiss Microscopy). Images were obtained as Z-stack sections of 0.2 mm per section containing 18 Z-stacks for each channel. A maximum projection file was generated to identify and count the number of foci and nuclei, and nuclei with >5 RAD51 foci per nucleus were counted as positive.

RAD51 foci from 100 nuclei were measured in three to seven independent experiments. Data were processed and plotted using GraphPad Prism 8.4. Standard deviations from the data were calculated and presented as error bars together with the mean values. $p$ values were calculated using one-way ANOVA and Tukey's multiple comparison test.

## Generation of multiple sequence alignment

Sequences of BRCA2 orthologs were retrieved from Uniprot[64] and manually curated; only sequences that contain at least one BRC repeat (Pfam ID 00634) and the OB-fold DNA binding region (Pfam ID 09169, 09103, and 019104), including the Tower domain (Pfam ID 09121) were selected for analysis. Sequence space was further narrowed by homology reduction to a defined number of sequences describing the four kingdoms of Eucarya (25 species), as well as the subkingdoms and classes Animalia (15 species), vertebrates (15 species), Mammalia (15 species), and primates (15 species) allowing the generation of 5 subsets. Sequences were evenly distributed across species in order to represent all branches of the tree of life equally. A total of 63 sequences were included in the analyses. Data were plotted using Microsoft Excel 16.

## Multiple sequence alignments

All sequences were aligned using Quickprobs 2[65]. A compiled file of all species (63 species in total) was aligned initially, and subsequently, all subsets composed as described above were aligned individually. Poor conservation resulted in substantially gapped alignments, resulting in the number of alignment positions on the x-axis greatly exceeding the number of amino acids found in BRCA2. The global alignment was analyzed for degree of conservation using the Jensen-Shannon divergence algorithm[66]. In order to proceed with local alignments scores for selected domains, i.e. the BRC4 repeat (1517–1548), the Tower domain (2846–2950), the first OB-fold (2718–2800), the CTRB (3260–3337) and a random disordered region (296–387), the local regions were aligned using Clustal 2.1[67] to obtain the percent identity matrix (PIM). Scores taken from the PIM were plotted as a function of species divergence obtained from timetree.org[68]. Data were plotted in Microsoft Excel 16.

## Disorder prediction

Protein disorder was predicted using three different algorithms IUPred2A[36], PrDOS[37], PONDR[38], and MFDp2[39], using different prediction principles for unbiased comparison. The result from all four algorithms were plotted as a function of BRCA2 sequence positions. For all three algorithms, a value of 1 describes the probability to find a fully folded protein segment and a value of −1 describes the probability to find a fully unfolded segment. Data were plotted in GraphPad Prism 8.4.

## Short linear motif prediction

CTRB fragment (3260–3337) was analyzed for human kinase motifs using the GPS web server[69], applying the subset of human kinases and setting the threshold to high. For ubiquitination and sumoylation, we likewise applied GPS-Uber and GPS-SUMO[70] using a medium threshold for both, site and motif identification.

## Circular dichroism

Far-UV CD measurements were conducted at 195–250 nm at a protein concentration of 10 μM of BRCA2 CTRB fragment (3260–3337) in a buffer containing 1x PBS, pH 6.0, 10 mM DTT, at 25 °C using a JASCO J810 spectropolarimeter and a 1 mm path length. Data were plotted with GraphPad Prism 8.4.

## Nuclear magnetic resonance spectroscopy

All NMR spectra were recorded on an Agilent DD2 800 MHz spectrometer using a room temperature probe and standard pulse programs from the Vnmrj BioPack. For an assignment, we prepared $^{15}$N-$^{13}$C-labeled BRCA2 CTRB-derived polypeptide consisting of amino acid residues 3260–3337 (20 μM) in a buffer containing 1 x PBS, pH 6.0, 10 mM DTT, and 10% D$_2$O (v/v), 125 μM 2,2-dimethyl-2-silanepentane-5-sulfonic acid at 5 °C. The backbone nuclei were assigned using a suite of HSQC, HNCA, HNCO, HN(CA)CO, HNCOCA, HNCACB, and CBCACONH[71–75]. It should be noted that the HNCACB and CBCACONH spectra were only used on a confirmatory basis where adequate, and no CB shifts were extracted for further analysis. The CA, CO, N, and NH chemical shifts have been deposited at the Biological Magnetic Resonance Bank (BMRB) under the accession number 51679. All 3D experiments were recorded using non-linear sampling with a 25% data reduction according to the Orekhov method incorporated into the Varian BioPack[76]. The assignment was completed for 95% of all non-proline residues. The sequence of the 12-mer ssDNA tested is given in Supplementary Table 1.

## NMR data processing and data analyses

The X-carrier frequency was determined by referencing to internal 2,2-dimethyl-2-silanepentane-5-sulfonic acid and indirectly for $^{15}$N and $^{13}$C dimensions using the conversion factors as described[77]. The spectra were processed using nmrDraw/nmrPipe[78] and qMDD[76]. The processed spectra were subject to CcpNmr Analysis[79]. Secondary chemical shifts were analyzed using the ssp software[43]. Chemical shift intensity perturbations in the absence and presence of an equimolar concentration of ssDNA (5-GGCTAT GCGTTA-3) were referenced to the intensity of the last residue in the sequence (N3337) calculated using Eq. 1:

$$\frac{I}{I_0} = \frac{(I_{\text{bound}}/I_{N3337,\text{bound}})}{(I_{\text{free}}/I_{N3337,\text{free}})} \qquad (1)$$

Where I and $I_0$ are the relative peak intensities in the presence and absence of ssDNA. Data were plotted using GraphPad Prism 8.4.

## Reporting summary

Further information on research design is available in the Nature Portfolio Reporting Summary linked to this article.

# Data availability

All data generated or analyzed during this study are included in the article and its Supplementary information. The NMR data generated in this study have been deposited in the Biological Magnetic Resonance Bank (BMRB) under accession number 51679. Source data are provided with this paper.

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

## Acknowledgements

This study was supported by research grants from the US National Institutes of Health (RO1 CA168635, RO1 ES007061, PO1 CA92584, R35 CA241801 (P.S.); R50 CA265315 (Y.K.); RO1 GM141091 and RO1 CA268641 (W.Z.); RO1 GM136717, RO1 CA23728, RO1 CA188347 (A.M.); R56 ES021454 (C.W.); R01CA246807 (S.B.), Danish Cancer Society (R167-A10921-B224) (C.S.), Cancer Prevention and Research Institute of Texas (CPRIT) (RP220269) (R.H.) and RP210102 (W.Z.), Congressionally Directed Medical Research Programs (BC191160) (A.M.), and a Gray Foundation Team Science Grant under the Basser Initiative (P.S.), ACS Postdoctoral Fellowship (PF-22-034-01-DMC) (C.M.R.), CPRIT Postdoctoral Fellowship (RP170345) (A.S.K.), and NIH predoctoral fellowship awards (F30CA260908, T32CA148724) (F.E.N.). P.S. is the holder of the Robert A. Welch Distinguished Chair in Chemistry (AQ-0012) and recipient of a Recruitment of Established Investigators Award from CPRIT (RR180029). A.M. is the holder of the Joe R. and Teresa Lozano Long Chair in Cancer and recipient of a Recruitment of Established Investigators Award from CPRIT (RR210023). S.B. is the holder of the Mays Family Foundation Distinguished Chair in Oncology.

## Author contributions

Y.K.: designed and conducted biochemical assays and wrote the manuscript. H.R.: conducted bioinformatic, biophysical, and CD analyses. V.P. and H.R.: conducted structural analysis by NMR. C.W., and P.Se: conducted DNA fiber and foci formation analyses. Z.H., A.K., W.Z., and A.B.: conducted cell-based HR assays and DNA damage sensitivity experiments. A.S.: contributed to domain mapping and mutant isolation. C.R., F.N., and J.K.: carried out in vitro DNA binding and pull-down assays. S.B., B.M., R.H., and A.M.: provided key research, materials, and participated in experimental design. M.J., L.L., and S.H. constructed plasmids. Y.K. C.W., H.R., C.S., and P.Su.: wrote the manuscript with contributions from all other authors.

## Competing interests

The authors declare no competing interests.
