## [Peer Review File · Nature Communications]

DNA Binding and RAD51 Engagement by the BRCA2
C-terminus Orchestrate DNA Repair and Replication Fork
PreservationREVIEWER COMMENTS

Reviewer #1 (Remarks to the Author):

BRCA2 is a very large protein that is predicted as mostly disordered. Despite its essential role in mitotic and meiotic HR, as well as in replication fork protection, its functional mechanisms are poorly described. In particular, very few BRCA2 disordered motifs responsible for integrating cellular signals and interacting with partners are yet identified. This article presents a series of convincing biochemical and biological data aiming at understanding the role of the BRCA2 disordered region encoded by exon 27 (also called CTRB) in HR and replication fork protection. From a molecular point of view, the authors identified RAD51 and DNA binding motifs in this BRCA2 CTRB region. From a biological point of view, they showed that this region facilitates loading of RAD51 on both ssDNA and RPA-coated ssDNA, and contributes to HR, resistance to MMC, HU and PARP inhibitor, assembly of RAD51 foci and protection of stressed replication forks. While I find this story new and exciting, I have a series of questions that should be addressed before publication.

In the section "Bioinformatic analyses revealed conserved motifs within BRCA2 CTRB", Figure 1: Whereas panels a et b are clear and provide information to the reader, panels c and d are difficult to interpret and should be moved to the supplementary data. Indeed, several tools exist for predicting disorder, but the tools used to produce these panels are not the most efficient ones (Nielsen & Mulder, 2019), and in particular in the case of BRCA2, they give results that are ambiguous. BRCA2 disorder propensity was already previously discussed. In particular, a plot showing the BRCA2 disorder propensity as calculated by SPOT-Disorder 2 (Hanson et al., 2019) was reported and, at least in part, experimentally confirmed using NMR (Julien et al., 2021).

Additional minor points:

The authors identified several conserved motifs in BRCA2 CTRB, which they described as:

- "two tandem PhePP motifs starting at F289 and F3298". The motifs are FVSP and FQPP, so writing that there are tandem PhePP motifs is not accurate.
- "a highly conserved cluster of positively charged residues spanning residues K3262 to R3269", is it the human numbering? If yes, then it is K3263.

In the section "Biophysical evidence for a DNA binding activity in the CTRB", a lot of experimental details are lacking. I understood from the Mat & Met that the NMR resonance assignment was performed on a 20 uM sample using a room temperature probe. It is complicate to acquire a HNCACB with a reasonable signal to noise ratio in these conditions (even using NUS). Can the authors give more details about how this has been carried out?

Also, did the authors carry a Size-Exclusion Chromatography experiment on the NMR sample before running the experiments, to be sure that it is homogeneous? Why did they record NMR spectra in the presence and absence of urea, as written in the Mat & Met? Which spectra were recorded in urea, which spectra were recorded without urea? Could the authors add in the legends of the CD and NMR figures in which conditions (buffer, pH, temperature, protein concentration) were the experiments performed? Could they show the whole HSQC spectrum in Fig. 2a?

Minor points:

- "Circular dichroism (CD) and 15N- Heteronuclear Single-Quantum Correlation Spectroscopy (HSQC)-NMR analyses revealed distinct spectra" -> why "distinct"?
- "and a 15N-HSQC NMR spectrum" -> the 15N-HSQC NMR spectrum, there is only one type of 15N-HSQC NMR spectrum.
- "an extraordinary extended nature of the region under investigation, only disrupted by a short stretch of residual helicity directly proceeding the second PhePP motif" -> there is the same pattern at the N-terminus of the protein fragment.
- "the CTRB region adopts an extended, almost modular conformation, in which the modularity could be mediated by conserved prolines in key positions." -> what do "modular" and "modularity" mean

here, as there is no folded structure?

In the section "Biochemical analysis of DNA binding by CTRB and mutant isolation", why only cluster A was studied, and not also cluster B (RSCGTK), while both clusters interact with DNA, as observed by NMR (Fig. 2b)?

In general, I cannot see how many replicates were performed in the case of the affinity pull down & other biochemical assays (ex: Fig 3d; also Fig 3e, Fig 3g in which there are error bars but I don't see how many experiments were used to calculate the error bars; same for Extended Fig 2b,c ...).

Minor points:

- In Extended Fig 2d, the authors show the impact of S3291A on RAD51 binding; did they try the phosphomimetic S3291E?
- For the purification of the MiniBRCA2 - DSS1 construct, I could not find on which protein was the HisTag.

Finally, I ran AlphaFold multimer, and using this program I could calculate 5 models of the complex between full length RAD51 and the BRCA2 CTRB peptide. Four of these models were highly similar and obtained high IDDT scores at the RAD51/BRCA2 interface. AlphaFold predicts that BRCA2 residues I3286 to T3306 interact with RAD51, with F3298 being at the centre of the binding site. S3291 does not contact RAD51. K3296 and R3302 are only very partially buried at the interface. The most buried residues from BRCA2 are T3288, V3290, A3294, F3298 and P3301. Based on this model, BRC4 and CTRB compete for RAD51 binding. Such tool could be used to support the biochemical analysis presented in this manuscript.

Reviewer #2 (Remarks to the Author):

Kwon and colleagues present compelling results on the function of the CTRB domain of BRCA2. The manuscript starts with a detailed analysis showing that this domain contains relatively well conserved features (comparable to other functional domain of BRCA2). Both in vitro and in vivo, the authors then show that this domain binds DNA and RAD51 and that both functions are required for HR and more specifically for RAD51's presynaptic filament assembly.

Overall, the manuscript is well written and data are clear and supportive of the conclusion. The manuscript provides important results on a domain currently not well understood and these data are important for BRCA2 biology and understanding its tumour suppressive function. I only have a few minor comments, see below, mostly on how the function of the CTRB domain relates to the other domains in BRCA2.

1. The authors claim that the CTRB contains tandem PhePP motifs. The second motif (most C-terminal) clearly shows a well-conserved PhePP sequence, but it is unclear to me why the authors claim the first sequence is also a PhePP motif, since the Phenylalanine is only partially conserved and a double PP does not seem to have co-evolved with this F.
2. In figure 4e the authors show no binding of mini-BRCA2_F/A to RAD51. Yet, this protein still contains BRC4 which is also known to bind RAD51. The authors claim that these results are consistent with the fact that CTRB associates with oligomeric RAD51, while BRC4 binds monomeric RAD51. It would be nice if the authors could back this conclusion up with more dedicated results.
3. More in general, I feel the authors could strengthen their manuscript by some more discussion on how the different domains of BRCA2 might work together. In my opinion, it is rather unexpected that the effects of the 4A and F/A mutant are so strong in the context of the mini-BRCA2 or full length protein (e.g. fig 4g, fig. 6bcd). What about the interplay between the CTRB and BRC-DBD domains?
4. In most assays the 4A mutant has a milder defect than the F/A mutant, except in the replication fork protection phenotype. Can the authors comment on this?

5. It would be nice if the authors could check whether the residues mutated in the 4A and F/A mutant are frequently occurring VUS or known pathogenic cancer mutations.
6. Minor detail: what do the * and ** mean in figure 5a?

Point-by-Point Response to the Critiques

Reviewer #1

Overall: The Reviewer found our work new and exciting but had a number of questions for us to address. Our response to these questions follows.

1. Figure 1: Whereas panels a and b are clear and provide information to the reader, panels c and d are difficult to interpret and should be moved to the supplementary data. Several tools exist for predicting disorder, but the tools used to produce these panels are not the most efficient ones, and in particular in the case of BRCA2, they give results that are ambiguous. BRCA2 disorder propensity was already previously discussed. In particular, a plot showing the BRCA2 disorder propensity as calculated by SPOT-Disorder 2 (Hanson et al., 2019) was reported and, at least in part, experimentally confirmed using NMR (Julien et al., 2021).

Response: These are valid points. Accordingly, we have removed Figure 1c and inserted the reference to the analysis of Hanson et al (2019; PMID 32173600) and Julien et al (2021; PMID 34356684). We have carried out testing of additional algorithms on the CTRB but could not find any that would match the quality of SPOT2. We have added one more recent and better performing algorithm (MFDp2 : which calculates per-residue prediction by applying an offset-correction via the inclusion of overall content predictors) to the prediction analysis panel, and moved the bioinformatic analysis to Supplementary Figure 1a. We agree that the different algorithms do not necessarily agree with one another. Therefore, we present a selection of algorithms to avoid introducing any bias that would agree better with our biophysical data.

2. Additional minor points: ...“two tandem PhePP motifs starting at F3289 and F3298”. The motifs are FVSP and FQPP, so writing that there are tandem PhePP motifs is not accurate.”; “..highly conserved cluster of positively charged residues spanning residues K3262 to R3269”, is it the human numbering? If yes, then it is K3263.

Response: We apologize for the confusing description and error. We now use the term “FXXP” instead of “PhePP” in the revised manuscript and have corrected the K3263 residue number.

3. In the section “Biophysical evidence for a DNA binding activity in the CTRB”, a lot of experimental details are lacking. I understood from the Materials & Methods that the NMR resonance assignment was performed on a 20 μ M sample using a room temperature probe. It is complicated to acquire a HNCACB with a reasonable signal to noise ratio in these conditions (even using NUS). Can the authors give more details about how this has been carried out?

Response: We apologize for not being more specific about the raw data that went into the assignment process and the further calculation of the ssp data. As pointed out, the HNCACB and CBCACONH spectra only showed very weak signals and we used these two spectra only on a confirmatory basis. Our assignments are mainly based on HNCOCA, HNCA and HNCO only. The few CB chemical shifts unambiguously assigned were not extracted. Neither were they deposited at the BMRB nor used for the ssp calculations. We have now clarified this in the Methods section.

4. Did the authors carry a Size-Exclusion Chromatography experiment on the NMR sample before running the experiments, to be sure that it is homogeneous?

Response: Because of the low yield of the isotope-labeled protein, we could not carry out size exclusion chromatography prior to NMR analysis. However, we are confident about the validity of our analyses and conclusions as any aggregated species would not contribute to the NMR spectra due to their size and hence the low tumbling time. In addition, we carried out buffer screening via DLS on an unlabeled sample to identify conditions where the sample was monodispersed.

5. Why did they record NMR spectra in the presence and absence of urea, as written in the Materials & Methods? Which spectra were recorded in urea, which spectra were recorded without urea?

Response: We apologize the mistake in the description. The spectra presented in the manuscript were obtained without urea. We used the urea spectrum for internal referencing in a different context, but the data were not used for this manuscript. The description in the Nuclear Magnetic Resonance spectroscopy section has been corrected accordingly.

6. Could the authors add in the legends of the CD and NMR figures in which conditions (buffer, pH, temperature, protein concentration) were the experiments performed?

Response: The experimental conditions have been added to the legends of Fig. 1 and Supplementary Fig. 1b & c.

7. Could they show the whole HSQC spectrum in Fig. 2a?

Response: The spectrum is in Supplementary Fig. 1c. We have added a sentence referring to the whole HSQC spectrum in Fig. 1c.

8. “Circular dichroism (CD) and ^{15}N - Heteronuclear Single-Quantum Correlation Spectroscopy (HSQC)-NMR analyses revealed distinct spectra” -> why “distinct”?

Response: We meant to point out that the spectra of CTRB show typical characteristics of a disordered protein. In the revised manuscript, the word ‘distinct’ has been removed.

9. “and a ^{15}N -HSQC NMR spectrum” -> the ^{15}N -HSQC NMR spectrum, there is only one type of ^{15}N -HSQC NMR spectrum.

Response: This has been corrected.

10. “an extraordinary extended nature of the region under investigation, only disrupted by a short stretch of residual helicity directly proceeding the second PhePP motif” -> there is the same pattern at the N-terminus of the protein fragment.

Response: We believe the first short stretch of helicity near the N-terminus might be overestimated due to the N-terminal His tag. The apparent helicity and a discussion point have been added in the revised manuscript under the section “**Biophysical evidence for a DNA binding activity in the CTRB**”.

11. “the CTRB region adopts an extended, almost modular conformation, in which the modularity could be mediated by conserved prolines in key positions.” -> what do “modular” and “modularity” mean here, as there is no folded structure?

Response: We have rephrased the sentence for clarification. “The CTRB region adopts an extended conformation where motifs are aligned like beads on a string, in which the modularity of the motifs appears to be controlled by conserved prolines in key positions.”

12. In the section “Biochemical analysis of DNA binding by CTRB and mutant isolation”, why only cluster A was studied, and not also cluster B (RSCGTK), while both clusters interact with DNA, as observed by NMR (Fig. 2b)?

Response: The Cluster A mutant CTRB-4A was the first mutant we made and turned out to be clearly impaired for DNA binding in concordance with the NMR analysis as we presented in the paper. Given that the CTRB-4A mutant retains full ability to interact with RAD51 and is therefore a most desirable separation-of-function mutant, we conducted all subsequent studies with this mutant.

We are only beginning to examine the role of Cluster B in DNA binding and this will constitute an integral component of the Ph.D. dissertation of Francisco Neal, who is an MD/PhD student in the Sung laboratory and one of the authors of this research paper.

13. In general, I cannot see how many replicates were performed in the case of the affinity pull down & other biochemical assays (ex: Fig 3d; also, Fig 3e, Fig 3g in which there are error bars but I don't see how many experiments were used to calculate the error bars; same for Extended Fig 2b,c ...).

Response: The details of statistical analyses are presented in Methods. We have added them to the figure legends and modified bar graphs to reveal all the data points. The pulldown experiments (Fig. 3d and Fig. 4e: in the revised manuscript, Fig. 2d and Fig. 3e) were done at least three separate times and the results were highly reproducible. Representative images of the results are presented in this paper.

14. In Extended Fig 2d, the authors show the impact of S3291A on RAD51 binding; did they try the phosphomimic S3291E?

Response: We presented the impact of CTRB S3291E in Supplementary Figure 2c, lane 9 and 10, showing the mutant retains residual interaction with RAD51. It is described in the Results section: “The pulldown analysis revealed that the CTRB-S3291A and CTRB-S3291E mutants both retained significant, albeit reduced affinity for RAD51 (Supplementary Fig. 2c).”

15. For the purification of the MiniBRCA2 - DSS1 construct, I could not find on which protein was the HisTag.

Response: The His6 tag is on miniBRCA2 and this information is now provided in the Methods section.

16. Finally, I ran AlphaFold multimer, and using this program I could calculate 5 models of the complex between full length RAD51 and the BRCA2 CTRB peptide. Four of these models were highly similar and obtained high IDDT scores at the RAD51/BRCA2 interface. AlphaFold predicts that BRCA2 residues I3286 to T3306 interact with RAD51, with F3298 being at the center of the binding site. S3291 does not contact RAD51. K3296 and R3302 are only very partially buried at the interface. The most buried residues from BRCA2 are T3288, V3290, A3294, F3298 and P3301. Based on this model, BRC4 and CTRB compete for RAD51 binding. Such tool could be used to support the biochemical analysis presented in this manuscript.

Response: We very much appreciate the Reviewer sharing with us the AlphaFold analysis. We have reproduced the AlphaFold result of CTRB with the RAD51 monomer, highlighting F3298 as a key residue for RAD51 interaction. Given that CTRB is known to associate with oligomeric RAD51 (Davis and Pellegrini, 2007, PMC2096194). we also attempted to model CTRB interaction with the RAD51 dimer and trimer. Under these conditions, CTRB does not make any significant contacts with RAD51.

We have not tested whether BRC4 and CTRB compete for RAD51 interaction but note that BRC4 can disrupt the oligomeric structure of RAD51 and interacts with the RAD51 monomer in solution (Davies et al, 2001, PMID: 11239456; Pellegrini et al, 2002, PMID: 12442171), whereas the CTRB associates with oligomeric RAD51 in solution (Davis and Pellegrini, 2007, PMC2096194). We propose that CTRB may support RAD51 assembly through functioning as a RAD51 reservoir at sites of presynaptic filament assembly, as we proposed in Figure 4b and 6b.

Reviewer #2

Overall: The Reviewer noted that our manuscript is well written and the study provides important results on a BRCA2 domain currently not well understood, and that our data are valuable for understanding the biology and tumor suppression function of BRCA2.

The Reviewer had a few questions regarding how the function of the CTRB domain relates to the other domains in BRCA2. Below, we document how each of the points raised has been addressed.

1. The authors claim that the CTRB contains tandem PhePP motifs. The second motif (most C-terminal) clearly shows a well-conserved PhePP sequence, but it is unclear to me why the authors claim the first sequence is also a PhePP motif, since the Phenylalanine is only partially conserved and a double PP does not seem to have co-evolved with this F.

Response: Thank you for catching this oversight of ours. Reviewer #1 also made the same point (see #2 of this reviewer) and, accordingly, we have changed PhePP to FXXP in the revised manuscript.

2. In figure 4e the authors show no binding of mini-BRCA2-F/A to RAD51. Yet, this protein still contains BRC4 which is also known to bind RAD51. The authors claim that these results are consistent with the fact that CTRB associates with oligomeric RAD51, while BRC4 binds monomeric RAD51. It would be nice if the authors could back this conclusion up with more dedicated results.

Response: The original figure showed a reduced amount of RAD51 is able to associate with mini-BRCA2-F/A in affinity pulldown (original Fig. 4e, lanes 4 & 5). This result is highly reproducible and in congruence with published studies showing that BRC4 disrupts the RAD51 oligomer and interacts with monomeric RAD51 in solution (Davies et al, 2001, PMID: 11239456; Pellegrini et al, 2002, PMID: 12442171), whereas the CTRB associates with oligomeric RAD51 in solution (Davis and Pellegrini, 2007, PMC2096194).

We note that since GST-DSS1 has a similar mobility in SDS-PAGE analysis, it might have rendered the reduced RAD51 signal difficult to discern in the original figure. Because of this, we have generated a new figure panel (Fig. 3e in the revised manuscript) showing better separation between GST-DSS1 and RAD51. We believe that this new figure panel (Fig. 3e, lanes 6, 9, and 12) shows clearly that mini-BRCA2-F/A retains residual ability to interact with RAD51. We have also modified the Results section to indicate that interaction between BRCA2-F/A with RAD51 occurs through BRC4.

3. More in general, I feel the authors could strengthen their manuscript by some more discussion on how the different domains of BRCA2 might work together. In my opinion, it is rather unexpected that the effects of the 4A and F/A mutants are so strong in the context of the mini-BRCA2 or full-length protein (e.g. fig 4g, fig. 6bcd). What about the interplay between the CTRB and BRC-DBD domains?

Response: Our biochemical and cellular analyses of the CTRB mutants (F/A, 4A and the double mutants) show that both the DNA and RAD51 binding attributes of CTRB make a significant contribution toward DNA damage repair, HR efficiency, and protection of stressed replication forks, and that the double mutant exhibits stronger defects biochemically and biologically. We note that our results are in congruence with published work showing that deletion of gene exon 27 that encodes the CTRB engenders severe biological phenotypes (Kass et al, PMC5093336; Siaud et al, PMC3240595; Moynahan and Jasin, PMID11239455). Importantly, our study provides a satisfactory mechanistic explanation as to the role of the CTRB in the aforementioned biological processes.

As for functional interplay between the CTRB and BRC-DBD domains, our biochemical data show clearly that combining both entities, as in the case of miniBRCA2, leads to a significant enhancement of efficacy in HR mediator activity. Thus, CTRB and BRC-DBD are not redundant functional modules in BRCA2 but, rather, they act in concert to enhance RAD51 presynaptic filament assembly. We note that our results are incongruence with the biological results of Siaud et al (PMC3240595) showing that the CTRB is important for the biological efficacy of BRCA2 derived polypeptides. Following the Reviewer's advice, we have expanded the discussion of this concept in the revised Discussion section.

4. In most assays the 4A mutant has a milder defect than the F/A mutant, except in the replication fork protection phenotype. Can the authors comment on this?

Response: The Reviewer highlight a most intriguing attribute of the CTRB, which is that its DNA binding activity fulfills a crucial role in the protection of stressed replication forks. As we so in the expanded Discussion section, we surmise that the ability of CTRB to bind both ssDNA and dsDNA allows it to engage the ss-ds DNA junction of regressed replication forks to facilitate RAD51 nucleoprotein filament assembly. This concept is presented in Figure 6b, and we have expanded the description of the concept in the Discussion.

5. It would be nice if the authors could check whether the residues mutated in the 4A and F/A mutants are frequently occurring VUS or known pathogenic cancer mutations.

Response: There are BRCA2 variants of unknown significance (VUS; see Table below) within the DNA binding and RAD51 interaction motifs in the CTRB. As we now state in the Discussion, the experimental systems that we have devised should allow us to ascertain the functional impact of these CTRB VUS in future.

Protein change	Condition(s)	Accession	dbSNP ID
K3267R	Hereditary breast ovarian cancer syndrome	VCV001517698	
K3267N	Hereditary cancer-predisposing syndrome Hereditary breast ovarian cancer syndrome	VCV000653282	rs1555289963
R3269G	Hereditary cancer-predisposing syndrome Hereditary breast ovarian cancer syndrome	VCV000491393	rs1555289964
R3269I	Hereditary breast ovarian cancer syndrome Hereditary cancer-predisposing syndrome	VCV001023314	rs80359243
R3269T	Hereditary breast ovarian cancer syndrome Hereditary cancer-predisposing syndrome not provided	VCV000231757	rs80359243
R3269K	Hereditary cancer-predisposing syndrome Hereditary breast ovarian cancer syndrome	VCV000052901	rs80359243
F3298I	Hereditary breast ovarian cancer syndrome	VCV001019349.4	rs2073052295

6. Minor detail: what do the * and ** mean in figure 5a?

Response: These symbols pertain to the new Figure 4a. Specifically, ‘*’ indicates that reactions were performed in the absence of the dsDNA trap, while ‘**’ denotes GST-DSS1 that migrates above RAD51 in the SDS-PAGE gel. An explanation of these symbols has been added to Figure 4a legend.

REVIEWERS' COMMENTS

Reviewer #1 (Remarks to the Author):

The authors have answered to my questions and remarks.

Minor points:

- Fig1a: the mark of RAD51 under BRC4 is shifted and thus unclear; legend: « des » -> « does »
- Supp Fig 1a: legend > Four different disorder prediction algorithms (and not three).

Reviewer #2 (Remarks to the Author):

The authors addressed all the points I raised and provided a satisfactory explanation. In my opinion, this work is suitable for publication in Nature Communications.

Point-by-Point Response to Critique #1

Reviewer #1

1. Fig1a: the mark of RAD51 under BRC4 is shifted and thus unclear.

Response: In the revised manuscript, we have added a line below the BRC repeats and connected the RAD51 mark to the line, in order to indicate that RAD51 interacts with these repeats.

2. legend: « des » -> « does »

Response: We have corrected the error in Fig. 1a legend.

3. Supp Fig 1a: legend > Four different disorder prediction algorithms (and not three).

Response: We have corrected the error in Supplementary Fig. 1a legend.

.